# Theoretical principles explain the structure of the insect head direction circuit

**Pau Vilimelis Aceituno[1]\*[†], Dominic Dall'Osto[1][†], Ioannis Pisokas[2]\*[‡]**

[1]Institute of Neuroinformatics, University of Zürich and ETH Zürich, Zurich, Switzerland; [2]School of Informatics, University of Edinburgh, Edinburgh, United Kingdom

**Abstract** To navigate their environment, insects need to keep track of their orientation. Previous work has shown that insects encode their head direction as a sinusoidal activity pattern around a ring of neurons arranged in an eight-column structure. However, it is unclear whether this sinusoidal encoding of head direction is just an evolutionary coincidence or if it offers a particular functional advantage. To address this question, we establish the basic mathematical requirements for direction encoding and show that it can be performed by many circuits, all with different activity patterns. Among these activity patterns, we prove that the sinusoidal one is the most noise-resilient, but only when coupled with a sinusoidal connectivity pattern between the encoding neurons. We compare this predicted optimal connectivity pattern with anatomical data from the head direction circuits of the locust and the fruit fly, finding that our theory agrees with experimental evidence. Furthermore, we demonstrate that our predicted circuit can emerge using Hebbian plasticity, implying that the neural connectivity does not need to be explicitly encoded in the genetic program of the insect but rather can emerge during development. Finally, we illustrate that in our theory, the consistent presence of the eight-column organisation of head direction circuits across multiple insect species is not a chance artefact but instead can be explained by basic evolutionary principles.

**\*For correspondence:**
pau@ini.uzh.ch (PVA);
pisokasi@janelia.hhmi.org (IP)

[†]These authors contributed equally to this work

**Present address:** [‡]Janelia Research Campus, Howard Hughes Medical Institute, Ashburn, United States

**Competing interest:** The authors declare that no competing interests exist.

## Editor's evaluation

This important work suggests that the observed cosine-like activity in the head direction circuit of insects not only subserves vector addition but also minimizes noise in the representation. The authors provide solid evidence using the locust and fruit fly connectomes. The work raises important theoretical questions about the organization of the navigation system and will be of interest to theoretical and experimental researchers studying navigation.

## Introduction

Insects exhibit an impressive ability to navigate the world, travelling long distances to migrate, find food or reach places of interest before returning to their nests (*Müller and Wehner, 1988*; *Menzel et al., 1996*; *Heinze et al., 2013*; *Collett, 2019*), a feat that requires them to keep track of their orientation across long journeys (*Mappes and Homberg, 2004*; *Merlin et al., 2012*; *Warren et al., 2019*; *Collett, 2019*; *Beetz et al., 2022*). This orientation tracking is achieved by the use of visual cues as well as integrating angular velocity signals over time to maintain a heading estimate relative to a starting angle (*Seelig and Jayaraman, 2015*; *Taube, 2007*), known as *heading integration*.

Electrophysiological and calcium imaging studies have shown that the neural population encoding the head direction in insects has a sinusoid-shaped activation pattern (*Labhart, 1988*; *Labhart, 2000*;

**eLife digest** Insects, including fruit flies and locusts, move throughout their environment to find food, interact with each other or escape danger. To navigate their surroundings, insects need to be able to keep track of their orientation. This tracking is achieved through visual cues and integrating information about their movements whilst flying so they know which direction their head is facing.

The set of neurons responsible for relaying information about the direction of the head (also known as heading) are connected together in a ring made up of eight columns of cells. Previous studies showed that the level of activity across this ring of neurons resembles a sinusoid shape: a smooth curve with one peak which encodes the animal's heading. Neurons downstream from this eight-column ring, which relay velocity information, also display this sinusoidal pattern of activation.

Aceituno, Dall'Osto and Pisokas wanted to understand whether this sinusoidal pattern was an evolutionary coincidence, or whether it offers a particular advantage to insects. To answer this question, they established the mathematical criteria required for neurons in the eight-column ring to encode information about the heading of the animal. This revealed that these conditions can be satisfied by many different patterns of activation, not just the sinusoidal shape.

However, Aceituno, Dall'Osto and Pisokas show that the sinusoidal shape is the most resilient to variations in neuronal activity which may impact the encoded information. Further experiments revealed that this resilience only occurred if neurons in the circuit were connected together in a certain pattern.

Aceituno, Dall'Osto and Pisokas then compared this circuit with experimental data from locusts and fruit flies and found that both insects exhibit the predicted connection pattern. They also discovered that animals do not have to be born with this neuronal connection pattern, but can develop it during their lifetime.

These findings provide fresh insights into how insects relay information about the direction of their head as they fly. They suggest that the structure of the neuronal circuit responsible for encoding head direction was not formed by chance but instead arose due to the evolutionary benefits it provided.

*Loesel and Homberg, 2001*; *Pfeiffer et al., 2005*; *Pfeiffer and Homberg, 2007*; *Kinoshita et al., 2007*; *Heinze et al., 2009*; *Homberg et al., 2011*; *El Jundi et al., 2014*; *El Jundi et al., 2019*). Furthermore, downstream neural populations also encode velocity signals as sinusoidal activations (*Lyu et al., 2022*).

Theoretical work has speculated that this recurring motif of sinusoidal activity patterns might be so prevalent because it enables easy elementwise vector addition, where vectors encoded as sinusoidal activity waveforms can be added together to give a sinusoidal waveform encoding the sum of the vectors (*Touretzky et al., 1993*; *Wittmann and Schwegler, 1995*; *Vickerstaff and Di Paolo, 2005*). This allows the encoded heading to be easily used by downstream circuitry to track the insect's position (*Mittelstaedt, 1985*; *Wittmann and Schwegler, 1995*; *Vickerstaff and Di Paolo, 2005*; *Haferlach et al., 2007*; *Wessnitzer et al., 2008*; *Sakura et al., 2008*; *Stone et al., 2017*). Further studies have shown that models closely aligned with biological data can indeed implement heading integration using sinusoidal activity patterns (*Pisokas et al., 2020*; *Turner-Evans et al., 2020*), and that such circuits can be learned (*Vafidis et al., 2022*).

Here, we show that enabling easy vector addition cannot be the unique driving factor for the presence of sinusoidal heading encodings, as many other circuits with different activity patterns can perform vector addition in the same way. This finding led us to question whether the sinusoidal activation patterns seen in insect navigation circuits are a coincidence, or if they might offer a particular functional advantage that was selected for during evolution.

To address this question, we consider the basic principles necessary for a circuit encoding direction. Of all the circuits fulfilling these requirements, the sinusoidal activity pattern offers the best resilience to noise for the encoded information. However, obtaining this activity requires a circuit with a specific connectivity pattern between neurons. Thus, our theory predicts that the heading integration circuit will have a sinusoidal connectivity pattern. We compare our predicted circuit with connectivity data for the desert locust (*Schistocerca gregaria*) and fruit fly (*Drosophila melanogaster*) using network analysis tools, showing a strong agreement. We then ask how an insect brain might develop such a

circuit, finding that a simple Hebbian learning rule is sufficient. Finally, we combine ideas from replication dynamics with our theory, which leads us to the conclusion that the eight-column structure is a consequence of basic theoretical principles, rather than an evolutionary coincidence.

## Results

### A theoretical circuit for heading integration

We consider a population of $N$ 'compass neurons' with an activity that encodes the direction of the insect as an angular variable $\theta$. We represent the activity of this population by a vector where each element corresponds to the activity of one neuron, $\mathbf{a}(\theta) = \left[\mathbf{a}_1(\theta), \mathbf{a}_2(\theta), ..., \mathbf{a}_N(\theta)\right]$. We take $N = 8$ neurons, consistent with data from many insect species which possess an eight-column organisation, with each column encoding a different direction (*Honkanen et al., 2019*; *Stone et al., 2017*; *Pisokas et al., 2020*).

Each neuron's activity is updated depending on its current firing rate and the inputs it receives, both from other neurons in the circuit and externally. We formulate this update rule as follows:

$$\dot{\mathbf{a}}(\theta) = -\mathbf{a}(\theta) + \phi \left[\mathbf{W}\mathbf{a}(\theta) + \mathbf{u}(t)\right], \tag{1}$$

where $\mathbf{W}$ is the circuit's weight matrix, representing the connections between neurons; $\phi$ is the neural activation function, that converts the total neural input into an output firing rate; and $\mathbf{u}(t)$ is the external input that encodes the insect's angular velocity, via the PEN population of neurons (*Green et al., 2017*; *Turner-Evans et al., 2017*; *Sayre et al., 2021*; *Pisokas et al., 2020*; *Turner-Evans et al., 2020*; *Hulse et al., 2021*).

To simplify our derivations we allow the neural activity values to be both positive and negative, interpreting these values as being relative to a baseline neural firing rate. Similarly, we allow the weights to be both positive and negative, a common simplification in computational models (*Li et al., 2023*; *Cornford et al., 2020*; *Kriegeskorte and Golan, 2019*). This simplification will be addressed in section 'Comparing the predicted circuit with biological data' where we compare our model with experimental data.

### Mathematical principles for neural heading integration

A circuit capable of performing heading integration must fulfil the requirements outlined in *Table 1*. The first two requirements allow us to establish the family of possible path integration circuits. We then use the principle of noise minimisation to determine which circuits perform best.

### Constraints on the neural activity

The neural activity should encode the insect's head direction with a matching topology. Because the heading is a single angular variable, the topology of the activity space should be a 1D circle.

Furthermore, the symmetry requirement implies that rotating the heading of the insect should rotate the neural activity vector without changing its shape. Concretely, whether the insect is facing north or east, the activity of the neural population as a whole should be the same, but with the identity of the neuron with each activity value being different.

**Table 1.** Requirements for a heading integration circuit.

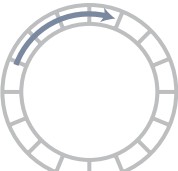 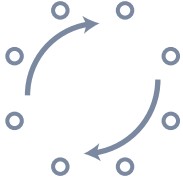 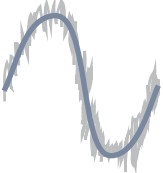

| Circular topology | Rotational symmetry | Noise minimisation |
| --- | --- | --- |
| The activity should have the same topology as the variable it is encoding to prevent discontinuities. To encode heading, the activity should have the topology of a 1D circle. | The heading integration circuit should work similarly, irrespective of the direction in which the insect travels. There should not be a bias for any direction. | The circuit should minimise the noise of the neural representation so the insect can navigate as precisely as possible. |

We can formalise the symmetry requirement by considering a head direction, $\theta$, and a rotation by an integer multiple, $k$, of the angular spacing between neurons, $\Delta\theta = \frac{2\pi k}{N}$. In this case, individual neuron activities follow the relation

$$\mathbf{a}_n(\theta + \Delta\theta) = \mathbf{a}_{n-k \mod N}(\theta),$$
(2)

which enforces that the neural activity vector is circularly rotated as the head direction changes. This relation can be expressed in the Fourier domain where, by the shift property, the circular rotation becomes a multiplication by a complex exponential:

$$\mathcal{F}_f\left[\mathbf{a}(\theta + \Delta\theta)\right] = \mathcal{F}_f\left[\mathbf{a}(\theta)\right] e^{\frac{i2\pi kf}{N}},$$
(3)

where $i = \sqrt{-1}$ and $f \in [0, ..., N-1]$ is the index of the spatial frequency, also called the harmonic. Activity with spatial frequency $f$ has $f$ 'bumps' around the ring.

Since the complex exponential has unit norm, the magnitude of the Fourier components remains the same for any rotation, $\|\mathcal{F}[\mathbf{a}(\theta + \Delta\theta)]\| = \|\mathcal{F}[\mathbf{a}(\theta)]\| \, \forall \Delta\theta$. Therefore, the shape of the activity pattern, $\mathbf{a} \equiv \mathbf{a}(0)$, can also be fully specified by its Fourier domain representation, $\mathcal{F}[\mathbf{a}]$, and the phase of this activity profile around the network encodes the heading of the insect:

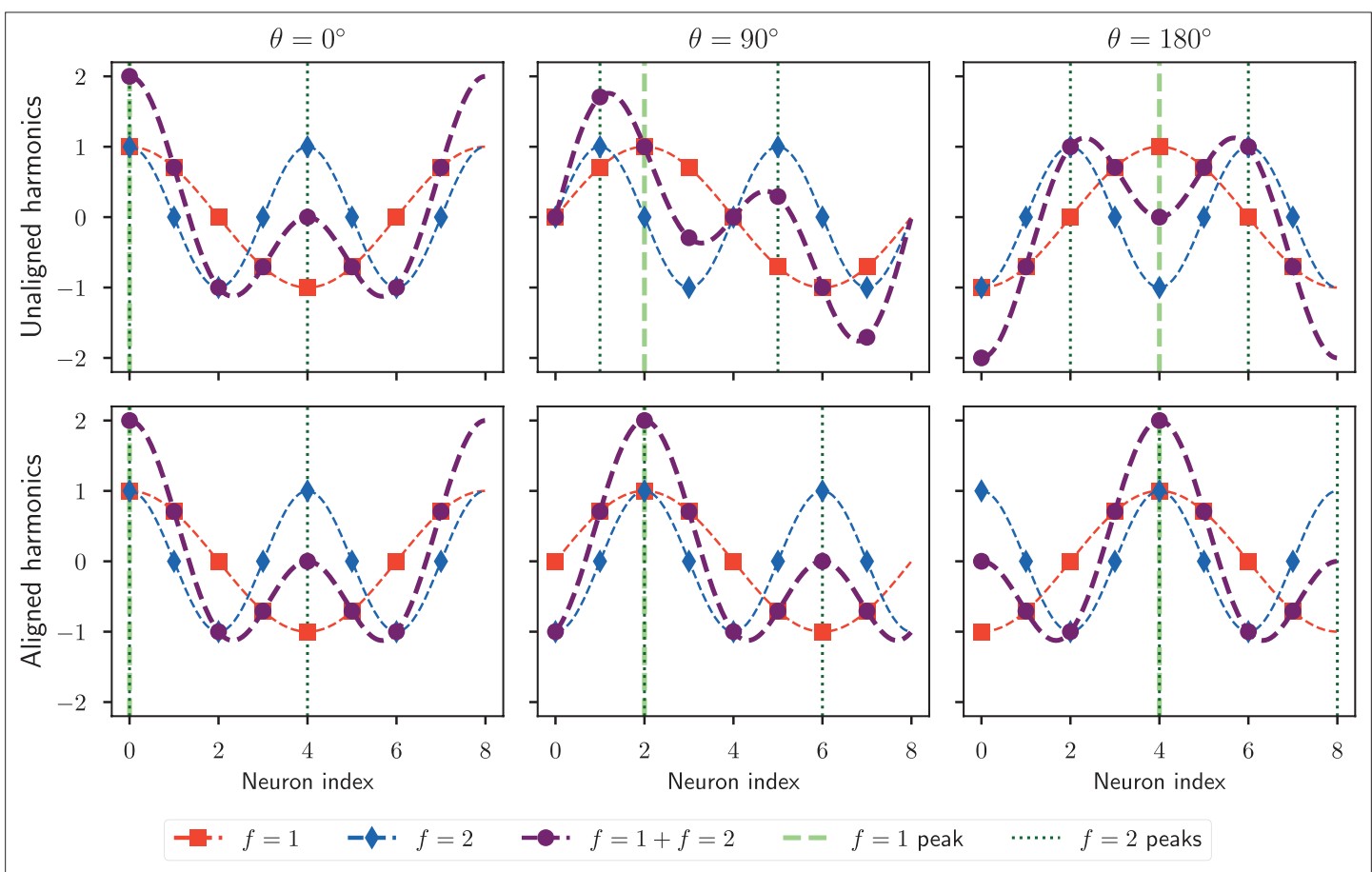

**Figure 1.** Encoding heading with multiple harmonics. Each panel shows the activity profiles encoding a particular heading value. Curves $f = 1$, $f = 2$, and $f = 1 + f = 2$ respectively denote the waveform of the first harmonic, second harmonic, and the sum of the two. The vertical dashed ($f = 1$ peak) and dotted lines ($f = 2$ peaks) indicate the neurons which respond maximally for the first and second harmonic, respectively. Top row: Encoding the heading as the sum of two independent harmonics causes the waveform to change shape as the insect rotates, because the waveform for each harmonic can only rotate a distance of $\frac{360°}{f}$ as the insect rotates a full revolution to ensure an unambiguous representation. Bottom row: If all harmonics are aligned to rotate at the same speed, the combined waveform shape does not change. However, this alignment implies that higher harmonic waveforms cannot be uniquely mapped back to a heading: here the $f = 2$ encoding is the same for $\theta = 0°$ and 180°.

$$\mathcal{F}_f\left[\mathbf{a}(\theta)\right] = \mathcal{F}_f\left[\mathbf{a}\right] e^{i\theta f}. \tag{4}$$

Taking the inverse Fourier transform with the constraint that the neural activities must be real, we get the following form for the neural activity,

$$\mathbf{a}_n(\theta) = \sum_{f=0}^{N-1} \|\mathcal{F}_f\left[\mathbf{a}\right]\| \cos\left(\frac{2\pi f n}{N} + \theta f\right). \tag{5}$$

It should be noted that the phase offset, $\theta f$, of each cosine waveform scales with the spatial frequency. This is because higher frequency waveforms have shorter wavelengths (in terms of number of neurons), so need their phases to rotate more quickly to move at the same speed around the network.

To explain this in more detail, we consider the case where this scaling is not used and the phase offset is the same for all harmonics, shown in *Figure 1* top row. For $f = 1$ (red line in *Figure 1*), the waveform has one bump and a wavelength of 8 neurons, so a 180° rotation corresponds to the waveform moving 4 neurons. But for $f = 2$ (blue line in *Figure 1*), the waveform has two bumps and a wavelength of 4 neurons, so a 180° rotation corresponds to moving only 2 neurons, while a 4 neuron shift would correspond to a full rotation. The fact that the waveforms for different frequencies move at different speeds through the network implies that, if multiple spatial frequencies are used to independently encode the heading, their positions would not be aligned and the combined waveform would change shape as different angles are encoded (purple line in *Figure 1*). This violates the rotational symmetry principle, as the activity would look significantly different when the insect is facing north compared to south.

To solve this problem of misaligned harmonics, we return to the case specified in *Equation 5* where the phase offset is scaled linearly with the spatial frequency so that all waveforms move at the same speed and rotational symmetry is ensured, which is shown in *Figure 1* bottom row. For a network with $N = 8$ neurons, a 180° rotation shifts the activity waveform by 4 neurons. This 4 neuron shift corresponds to a $\frac{4}{8}$ or 180° phase offset for the $f = 1$ waveform, which has a wavelength of 8 neurons, but a $\frac{4}{4}$ or 360° phase offset for the $f = 2$ waveform, which has a wavelength of 4 neurons. While this maintains rotational symmetry, it implies that the $f = 2$ waveform is the same whether the heading angle 0° or 180° is encoded. As a consequence, a higher harmonic waveform (having $f > 1$) cannot on its own specify the encoded angle. The activity in all harmonics must be considered simultaneously to decode the angle, but even then this decoding might not be unique in the presence of noise (see section 'Ambiguities in multiple harmonics decoding with drift'). The difficulties associated with encodings utilising multiple harmonics will be further addressed in the following sections.

## Constraints on heading integration circuits

The basic assumptions outlined earlier also constrain the possible heading integration circuits – such circuits should allow for neural activity with the required topology and rotational symmetry to stably exist and propagate. Together, these two principles require that the activity in the circuit should have a constant total magnitude. We consider that this constraint is enforced by the nonlinear neural activation function, $\phi$, in *Equation 1*, as detailed in section 'Path integration dynamics for heading and position'.

If the network activity is at the desired level, and the external input $\mathbf{u}(t)$ is projected onto the ring attractor such that it does not alter the total network activity, then we can consider the network dynamics to be linear at this operating point. Therefore, while within the space of possible activities that corresponds to the ring attractor, our circuit dynamics are effectively described as,

$$\dot{\mathbf{a}}(\theta) = -\mathbf{a}(\theta) + \mathbf{W}\mathbf{a}(\theta) + \mathbf{u}(t). \tag{6}$$

The rotational symmetry principle also applies to the network. For the same shaped activity waveform to be able to stably exist at any position around the network, the network connectivity should also be rotationally symmetric. For example, the connection strength between the neurons encoding the north and north-east directions should be the same as between those encoding south and south-west. Mathematically, this imposes that the weight matrix, $\mathbf{W}$, is circulant, specifically that $\mathbf{W}_{n,m} = \mathbf{W}_{n+k,m+k}$.

This matrix is fully specified by its first row, called the *connectivity profile* and denoted $\omega$, meaning that we can express the product of the matrix with the neural activity as

$$\left(\mathbf{W}\mathbf{a}(\theta)\right)_k = \mathbf{W}_k\mathbf{a}(\theta) = \sum_{m=0}^{N-1} \mathbf{W}_{k,m}\mathbf{a}_m(\theta) = \sum_{m=0}^{N-1} \mathbf{W}_{0,m-k}\mathbf{a}_m(\theta) = \sum_{m=0}^{N-1} \omega_{m-k}\mathbf{a}_m(\theta), \tag{7}$$

which can be simplified in terms of the convolution operation,

$$\mathbf{W}\mathbf{a}(\theta) = \omega * \mathbf{a}(\theta). \tag{8}$$

Considering the case where the insect is not moving, $\mathbf{u}(t) = 0$, and the network activity is stable, $\dot{\mathbf{a}}(\theta) = 0$, we can combine *Equation 6* and *Equation 8* to get a relation for the stable network activity

$$\mathbf{a}(\theta) = \omega * \mathbf{a}(\theta). \tag{9}$$

In the Fourier domain, this simplifies into

$$\mathcal{F}_f\left[\mathbf{a}(\theta)\right] = \mathcal{F}_f\left[\omega\right]\mathcal{F}_f\left[\mathbf{a}(\theta)\right]. \tag{10}$$

As for the activity waveform analysis, we note that the Fourier transform is taken on the neural indices, not on the temporal domain.

Here, we have $N$ equations that have to be satisfied, since the Fourier transform of an $N$-dimensional vector is also $N$-dimensional. For each harmonic frequency $f$, *Equation 10* has two solutions:

- $\mathcal{F}_f\left[\omega\right] = 1$, which implies that activity with this spatial frequency is stable in the network, and therefore that it can encode the insect's heading.
- $\mathcal{F}_f\left[\mathbf{a}(\theta)\right] = 0$, meaning that the activity in this harmonic is zero, and thus nothing can be encoded in this frequency.

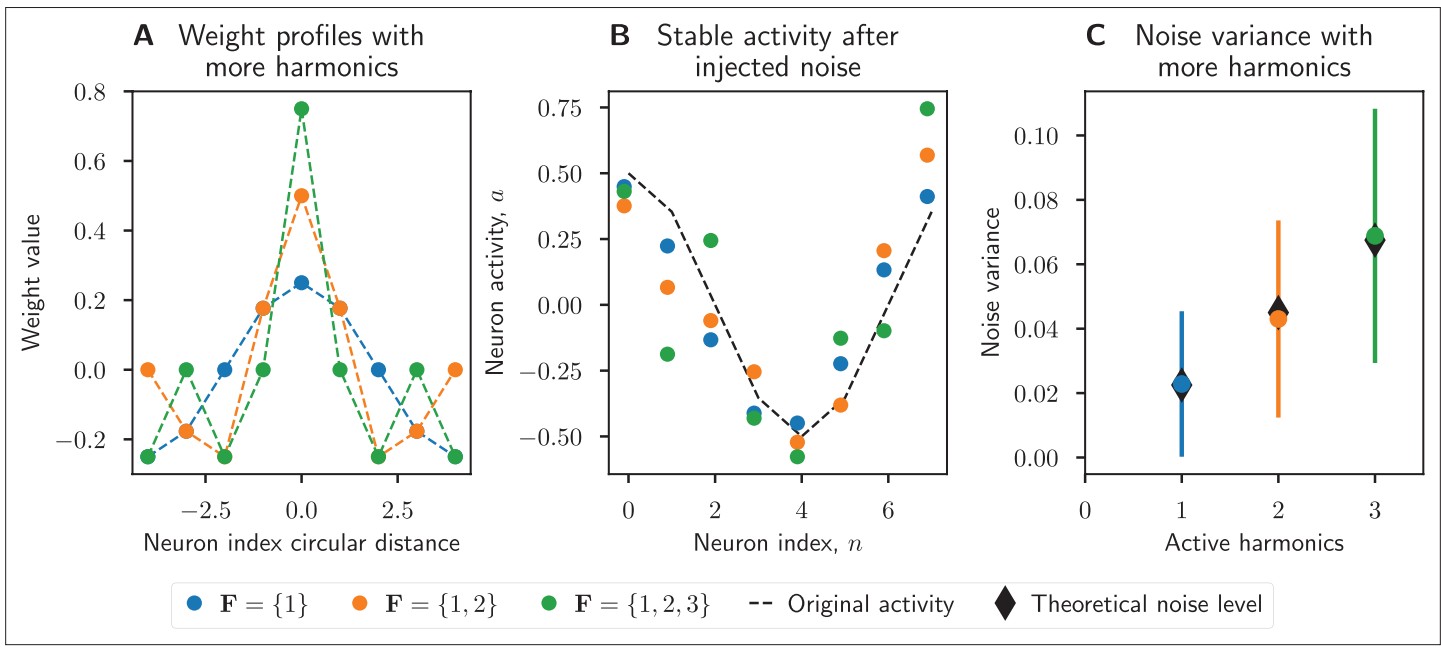

**Figure 2.** Increased number of harmonics introduces more noise. Increasing the number of active harmonic frequencies increases the effect of errors in the network. (**A**) Weight matrix profiles, $\omega$, for networks with increasing numbers of harmonics. (**B**) Normally distributed noise with zero mean and standard deviation 0.3 was added to the network activity, then the network state updated until it reached steady state. Networks with fewer harmonics better filtered out noise. (**C**) Noise variance increases linearly as the number of active harmonics increases, as predicted by *Equation 12*. The sample size was n=1000 trials for each active harmonics set, and the error bars show the standard deviation over trials.

## Minimising noise propagation in the circuit

The only constraint on the connectivity weights given by *Equation 10* is that, if a frequency is used for encoding then $\mathcal{F}_f[\omega] = 1$. There is no restriction on the weights for the inactive harmonics – they remain free parameters.

However, non-encoding channels can still propagate noise, which would be prevented by setting $\mathcal{F}_f[\omega] = 0$. To illustrate this, we consider white noise, denoted by $\epsilon$, that is added to the neural activity. When this noisy activity evolves in accordance with the dynamics from *Equation 8*, we have

$$\omega * \big(\mathbf{a}(\theta) + \epsilon\big) = \omega * \mathbf{a}(\theta) + \omega * \epsilon, \tag{11}$$

where the term $\omega * \epsilon$ corresponds to noise and should therefore be dampened. We want to minimise the strength of that noise, which is quantified by its variance

$$\mathbf{Var}\left[\omega * \epsilon\right] = \mathbf{E}\left[\|\omega * \epsilon\|^2\right] = \|\omega\|^2 \mathbf{Var}\left[\epsilon\right]. \tag{12}$$

Hence the magnitude of the noise that passes from one time interval to the next is modulated by the magnitude of the weight vector. By Parseval's theorem,

$$\sum_f \left|\mathcal{F}_f[\omega]\right|^2 = \sum_n |\omega_n|^2 = \|\omega\|^2, \tag{13}$$

which implies that to minimise noise propagation in the network we should impose $\mathcal{F}_f[\omega] = 0$ for all harmonics, $f$, where $\mathcal{F}_f[\mathbf{a}(\theta)] = 0$. We show this in simulation in *Figure 2*, where only the first harmonic encodes information and we vary the weights of the other harmonics. The noise is indeed minimised if the weights for all non-encoding harmonics are set to zero.

This result establishes that all non-encoding harmonics in the network should be set to 0 to minimise noise propagation, and allows us to recover the circuit connectivity

$$\omega_n = \sum_{f \in \mathbf{F}} \cos\left(\frac{2\pi n}{N} f\right), \tag{14}$$

where $\mathbf{F}$ is the set of harmonics used to encode the head direction, $\mathbf{F} = \{f \in [1...N-1] : \|\mathcal{F}_f[\mathbf{a}]\| \neq 0\}$. The choice of $\mathbf{F}$ therefore determines both the harmonics used for encoding the angle in the activity and the connectivity of the circuit that supports this activity.

We leverage the same logic to prove that the number of encoding channels does not affect the signal-to-noise ratio in the network. If we have $c$ encoding channels, each with the same activity, the total activity will grow linearly with $c$. The noise will also grow with $\|\omega\|^2 = c$, meaning that the signal-to-noise ratio will remain constant.

We now return to the question of whether using one or multiple harmonics is better. As mentioned, the signal-to-noise ratio in the network remains constant as additional encoding channels, $c$, are used. But additional channels imply additional harmonics, which by Parseval's theorem require a higher total neural activity. If the total activity in the network is limited, the signal-to-noise ratio is in fact *decreased* when using more than one harmonic.

$$\mathrm{SNR} \propto \frac{\|\mathbf{a}\|^2}{\|\omega\|^2} = \frac{\|\mathbf{a}\|^2}{c}. \tag{15}$$

Additionally, as detailed in sections 'Constraints on the neural activity' and 'Ambiguities in multiple harmonics decoding with drift', circuits using multiple harmonics perform worse in high noise environments because they are not guaranteed to provide an unambiguous orientation encoding. Finally, as we will discuss in sections 'Learning rules and development' and 'Convergence of Oja's rule with multiple harmonics', the use of multiple harmonics also complicates the circuit's development. We therefore only select circuits that use a single harmonic for further analysis, reducing the number of possibilities to $N$.

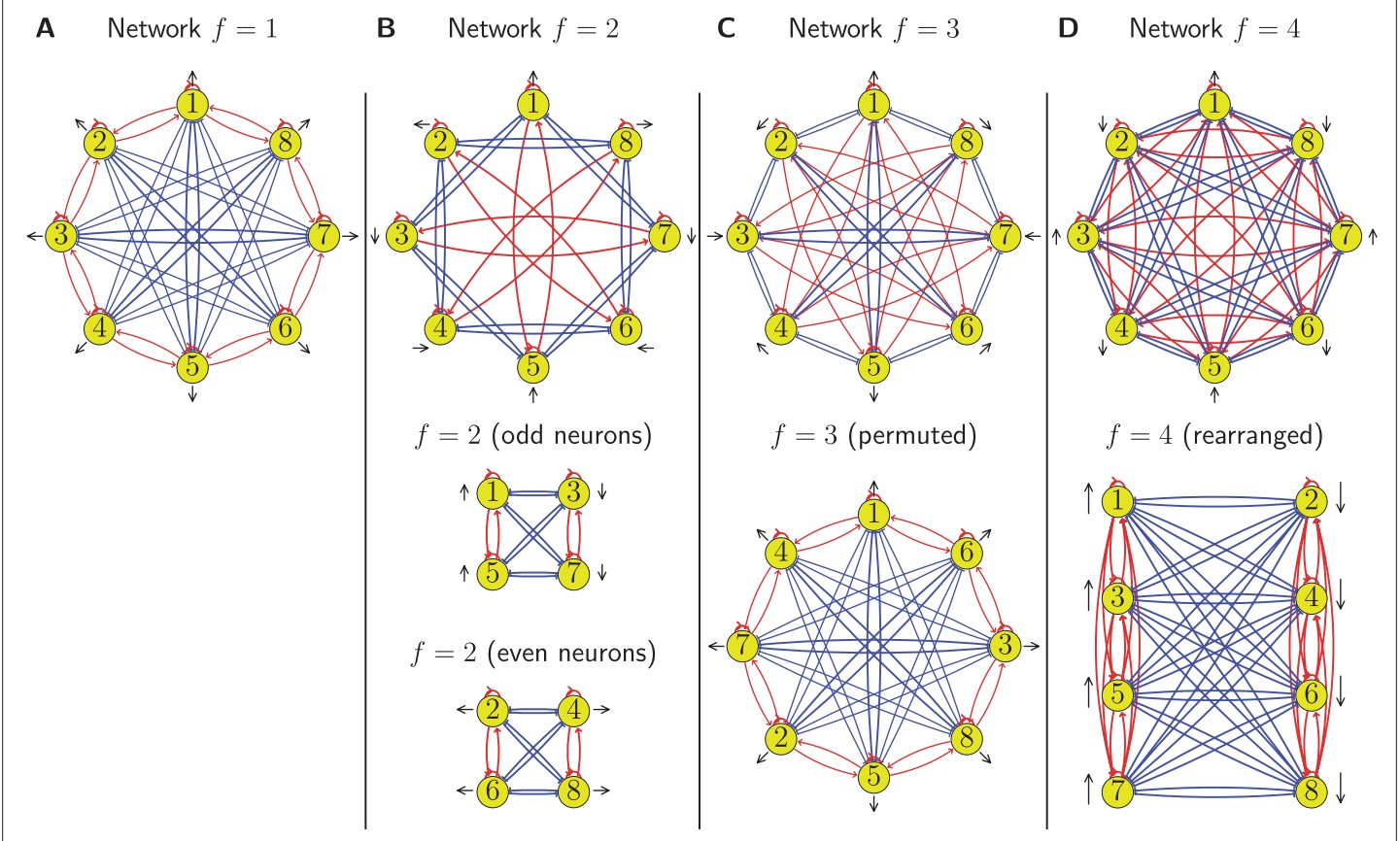

**Figure 3.** Circuits for encoding with different harmonics. We show four circuits corresponding to each of the individual harmonics $f = 1, 2, 3, 4$ in panels (**A**, **B**, **C**, and **D**), respectively. Excitatory synapses are marked in red and inhibitory in blue. Neurons are shown in yellow with each having a black arrow that marks the direction to which it is tuned from **Equation 16**. The $f = 1$ circuit is the simplest and constitutes our baseline. For the other cases we plot the original connectivity in the upper row and the rearranged network in the lower row. We find that the $f = 2$ circuit consists of two independent subnetworks encoding orthogonal directions, the $f = 3$ case is identical to $f = 1$ after permuting neuron indices, and the $f = 4$ case results in two connected groups of neurons inhibiting each other, hence it can only encode one direction. As such, all cases either have a degenerate ring structure ($f = 2, 4$) or are equivalent to $f = 1$ after permutation.

## Determining the optimal circuit

As we consider networks with $N = 8$ neurons, consistent with multiple insect species (**Honkanen et al., 2019**; **Stone et al., 2017**; **Pisokas et al., 2020**), the possible single harmonic circuits are $f = \{1, 2, 3, 4, 5, 6, 7\}$, where the zeroth harmonic is discarded because it only represents the baseline neural activity. This gives the following activities and weights, derived from **Equation 5** and **Equation 14**

$$\mathbf{a}_n(\theta) = \cos\left(\frac{2\pi n f}{N} + \theta\right) \quad \omega_n = \cos\left(\frac{2\pi f n}{N}\right). \tag{16}$$

The circuits for the first four harmonics are plotted in **Figure 3**, but not all of these circuits are valid. In particular, circuits with even spatial frequencies have multiple neurons with identical activity values because they share a common divisor with $N = 8$, as explained in detail in section 'Degenerate circuits'.

For $f = 4$ there are only two unique activity values, $\mathbf{a}_n = \pm \cos(\theta)\ \forall n$, so this circuit can only encode one dimension, not a circular topology (see **Figure 3D**). For $f = 2$ and $f = 6$ (not plotted), there are four unique activity values, allowing the angle to be properly encoded. However, these circuits are degenerate because they are composed of two independent subcircuits, each encoding one direction (see **Figure 3B**). Because the weights connecting the two subcircuits are all zero, the activity in each

subcircuit is independent of the activity in the other, and so the overall activity cannot be constrained to have the required circular topology.

All circuits with odd harmonic frequencies are equivalent. For example, as shown in *Figure 3*, the connections in the circuit for $f = 3$ are the same as those for the $f = 1$ circuit after the neuron identities are permuted. As detailed in section 'Equivalence under permutation', the frequencies $f = \{1, 3, 5, 7\}$ always give the same circuit because the odd frequencies are coprime with the number of neurons $N = 8$.

Since the activities and weights of all the non-degenerate circuits $f = \{1, 3, 5, 7\}$ are the same as the base harmonic $f = 1$ up to a permutation, we choose the lowest harmonic $f = 1$, which gives us the following activity and weights:

$$\mathbf{a}_n(\theta) = \cos\left(\frac{2\pi n}{N} + \theta\right)$$
$$\omega_n = \cos\left(\frac{2\pi n}{N}\right).$$

(17)

## Comparing the predicted circuit with biological data

Our theory proposes an optimally noise-resistant circuit for heading integration, and its corresponding activity. The prediction that heading should be encoded as a sinusoidal activity bump is consistent with previous theoretical models (*Touretzky et al., 1993*; *Wittmann and Schwegler, 1995*; *Hartmann and Wehner, 1995*; *Zhang, 1996*; *Vickerstaff and Di Paolo, 2005*; *Haferlach et al., 2007*; *Stone et al., 2017*), as well as experimental evidence in both the locust and fruit fly (*Heinze et al., 2009*; *Turner-Evans et al., 2017*). We note, however, that data from the fruit fly shows a more concentrated activity bump than what would be expected from a perfect sinusoidal profile (*Seelig and Jayaraman, 2015*; *Turner-Evans et al., 2017*), and that calcium imaging (which was used to measure the activity) can introduce biases in the activity measurements (*Siegle et al., 2021*; *Huang et al., 2021*). Thus the sinusoidal activity we model is an approximation of the true biological process rather than a perfect description.

Importantly, our theory proposes that the optimally noise-resilient heading integration circuit should have synaptic weights that follow a sinusoidal pattern, even though such weights are not necessary for producing sinusoidal activity as discussed in section 'Minimising noise propagation in the circuit'. This is the main prediction of our theory and we sought to validate it using connectivity data from the locust and fruit fly.

However, before we can directly compare our model with biological circuits, we must address a number of modelling simplifications:

- In the model we considered a population of 8 neurons, but in insects there are eight neural columns, each with several neuron types (EPG, PEG, PEN, Delta7). Of these we model only the EPG (known as the compass neurons) which encode the integrated head direction.
- The synaptic connections between neurons in our model can be both positive and negative, while biological neurons follow Dale's law, meaning that a single neuron can either have only positive or only negative synapses.
- The neural activity in the model was centred around zero and could have both positive and negative activity (firing rates), while real neurons only have positive firing rates.

Therefore, we simplified the biological connectivity to produce an equivalent circuit that could be directly compared with our model prediction. The neural population in our theoretical model corresponds to the biological EPG neurons, as these encode the integrated heading. We considered the other 3 neuron types that are part of the compass circuit (PEG, PEN, and Delta7 – see *Kakaria and de Bivort, 2017*; *Pisokas et al., 2020*) as just implementing connections between EPG neurons in accordance with biological constraints. We counted the number of different paths between EPG neurons, accounting for the sign of the connections (whether the path passed through an inhibitory Delta7 neuron), and used the net path count as a proxy for connectivity strength. This process is explained in detail in section 'Path counting'.

We then computed the average connectivity profile, i.e., how each neuron connected to its neighbours around the ring, and compared this profile to the closest fitting sinusoid. Because the neuron gains, absolute synaptic strength and biophysical properties of the neurons are unknown, the units of

the net path count are not necessarily equivalent to our abstract connection strength. We therefore fit an arbitrarily scaled and shifted sinusoid to the connection counts:

$$\omega_{m-n} = \beta \cos\left(\frac{2\pi(m-n)}{N}\right) + \gamma, \tag{18}$$

where $m - n$ is the circular distance between 2 neurons while $\beta$ and $\gamma$ are constants that are fit to minimise the precision-weighted mean squared error compared with the experimental connectivity profile (see section 'Fitting weights').

Analysing the data from *Pisokas et al., 2020*, we consider the shortest excitatory and inhibitory paths between EPG (also known as CL1a) neurons in the locust, which have lengths of 2 and 3, respectively. There are no direct connections of length 1, and all the paths of length ≥4 must pass through the same neuron-type multiple times. This path counting analysis for the locust is shown in *Figure 4A*, and the procedure is detailed in section 'Path counting'. We find that the connectivity profile between neurons for the locust is very close to sinusoidal in shape, supporting our theoretical prediction.

For the fruit fly, we used data from *Scheffer et al., 2020*, which provides synapse counts between pairs of neurons. These synapse counts were used as a proxy for connectivity strength, a view that has been validated in previous experiments (*Liu et al., 2022*; *Barnes et al., 2022*). After identifying the neurons and connections of interest, we grouped the neurons in eight columns following the logic presented in *Pisokas et al., 2020*, with detailed methodology explained in section 'Data preprocessing'. We repeated the path counting analysis for the fruit fly with synapse count data (*Figure 4B*), and found that while the data are noisy, the connectivity profile fits a single sinusoid pattern reasonably well. However, the high variability in the synapse counts makes our hypothesis difficult to differentiate from alternative shapes (see section 'Fitting weights').

Taken together, this analysis shows that our theory is consistent with experimental data – using binary connectivity data for the desert locust and synapse count data for the fruit fly.

## Learning rules and development

Having validated the connectivity of our theoretical circuit by comparing it to experimental data, we ask whether our circuit lends itself to biological development. Specifically, we show that even though our circuit requires precise connection strengths, this connectivity can be developed naturally by a Hebbian learning rule. Our approach follows from previous research which has shown that simple Hebbian learning rules can lead to the emergence of circular line attractors in large neural populations (*Stringer et al., 2002*), and that a head direction circuit can emerge from a predictive rule (*Vafidis et al., 2022*). In contrast to this work, we focus only on the self-sustaining nature of the heading integration circuit in insects and show that our proposed sinusoidal connectivity profile can emerge naturally.

Because the weight matrix is circulant, its eigenvalues are equal to the Fourier spectrum of its first row, which we derived to only have one nonzero value, for $f = 1$. The network therefore only has a single eigenvalue and so projects activity into a single dimension. This operation is similar to classical dimensionality reduction methods such as principal component analysis, which can be implemented by Hebbian-like learning rules (*Dayan and Abbott, 2001*). We thus analyse the effects of incorporating a modified Oja's rule into our model, a classical variant of Hebbian learning where the synaptic strength between 2 neurons grows when both neurons are active simultaneously, and the total synaptic strength is regularised to prevent exploding weight growth,

$$\frac{d\mathbf{W}_{n,m}}{dt} = \eta\frac{d\theta}{dt}\left(a_m(t)a_n(t) - a_n^2(t)\mathbf{W}_{n,m}\right), \tag{19}$$

where $\mathbf{W}_{n,m}$ is the synaptic connection strength from neuron $m$ to neuron $n$, $a_m$ and $a_n$ are the pre- and post-synaptic activities, respectively, and $\eta\frac{d\theta}{dt}$ is an adaptive learning rate where the plasticity is proportional to the rotational speed, whereas in the classical Oja's rule it would be constant.

For our analysis we assume that the insect faces all possible directions, and therefore that the neural activity goes around the full circle. We then integrate the weight updates over some long period of time,

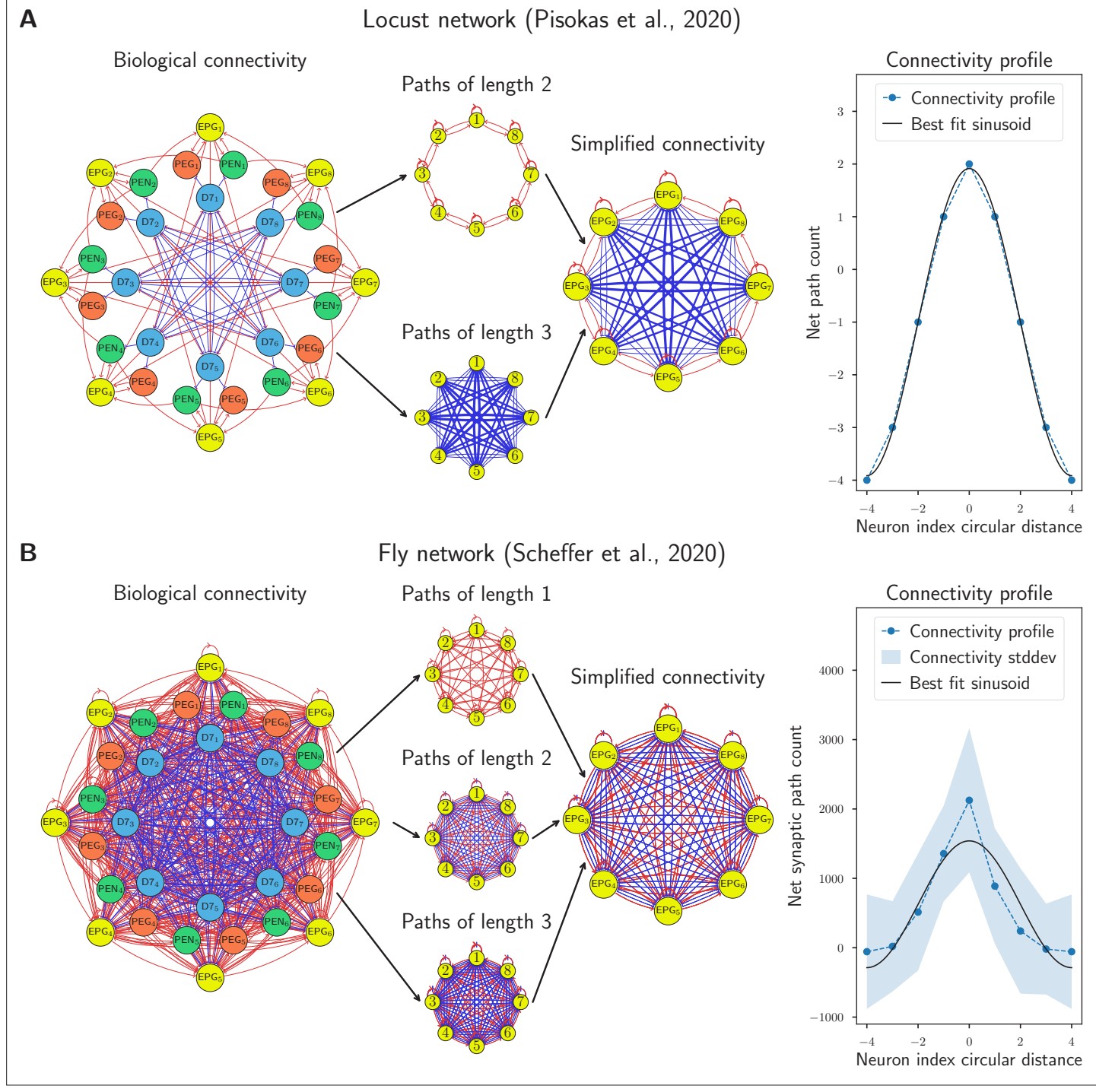

**Figure 4.** Comparison to biological data. The biological network models of (**A**) the locust and (**B**) the fruit fly, with four distinct neural populations each were simplified to equivalent networks with one population by counting paths of lengths 1, 2, and 3 between EPG neurons (which encode the integrated heading) and using the net signed path count as a proxy for connectivity strength. The average connectivity profile for each neuron to its neighbours around the ring was compared to the sinusoidal connectivity predicted by our theory. The locust network has no standard deviation because the data in *Pisokas et al., 2020*, are based on light microscopy which couldn't resolve variations between columns. Excitatory connections are shown in red and inhibitory connections in blue, while the strength of a connection is indicated by the line width.

$$\Delta \mathbf{W}_{n,m} = \int \frac{d\mathbf{W}_{n,m}}{dt} dt = \int \frac{d\mathbf{W}_{n,m}}{dt} \left(\frac{d\theta}{dt}\right)^{-1} d\theta = \eta \int_{\theta=0}^{2\pi} \left[ a_m(\theta)a_n(\theta) - a_n(\theta)^2 \mathbf{W}_{n,m} \right] d\theta, \quad (20)$$

where $\theta$ is the integration space. Applying the activity from *Equation 17*, we can find the fixed point of this update rule, when $\Delta \mathbf{W}_{n,m} = 0$:

$$\mathbf{W}_{n,m} = \frac{\eta \int_\theta \cos\left(\theta + \frac{2\pi n}{N}\right)\cos\left(\theta + \frac{2\pi m}{N}\right) d\theta}{\eta \int_\theta \cos^2\left(\theta + \frac{2\pi n}{N}\right) d\theta} = \cos\left(\frac{2\pi(n-m)}{N}\right). \quad (21)$$

Combined with *Equation 7* and *Equation 17*, this result means that if there is sinusoidal activity in the network, the weights will naturally converge to the optimal sinusoidal values by way of our variant of Oja's rule. This leads to the emergence of sinusoidal activity and weights from self-consistency: initial noisy sinusoidal activity will enforce weights that are close to a sinusoid, and those weights will filter out noise to make the activity even closer to a sinusoid, which will in turn make the weights more sinusoidal. This process will iteratively make the activity and the weights converge to the solution from *Equation 17*.

An important point in the use of Oja's rule is that it would tend to concentrate the activity in a single harmonic. In a linear network, the harmonics would compete during learning, leading to one single harmonic emerging and all others being suppressed, as shown in section 'Linear neurons'. For neurons with a nonlinear activation function, secondary harmonics would emerge, but would remain small under mild assumptions, as shown in section 'Neurons with nonlinear activations'. Oja's rule will still cause the weights to converge to approximately sinusoidal connectivity.

The finding that the weights will converge to a sinusoidal connectivity with learning has two interesting consequences from a biological standpoint: First, our circuit can emerge even when starting

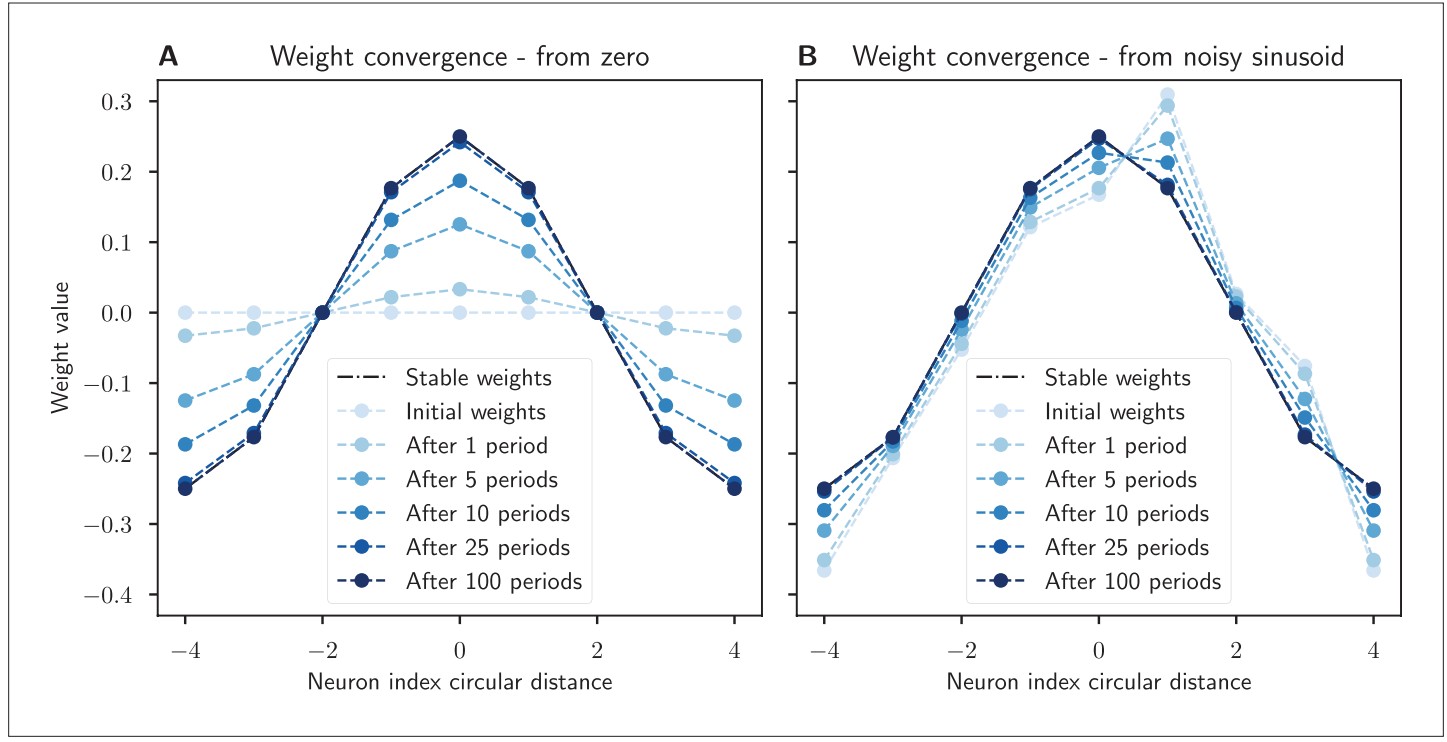

**Figure 5.** Recovery of synaptic weights with a Hebbian learning rule. The synaptic weights converge to a sinusoidal pattern under our modified Oja's rule when the network activity is dynamic and a sinusoidal input is provided. (**A**) The weights start at zero and slowly converge to the prescribed sinusoidal profile, showing that this connectivity can emerge from scratch. (**B**) The sinusoidal weights are perturbed by noise but learning ensures that the weight profile is corrected. In both cases the network's initial activity is corrupted with zero mean Gaussian noise. Noisy sinusoidal input is provided to rotate the activity bump around the network at a constant speed of 1/8 neurons per timestep. The simulation runs for 100 periods. Parameters: $N = 8$, integration timestep $\Delta t = 0.01$, $\eta = 0.1$, $\|\mathbf{a}\| = 1$, $\sigma_W = 0.2$, $\sigma_a = 0.2$, $\sigma_u = 0.2$.

with only very coarse initial weights, without the need for high precision initial connectivity. Second, this simple plasticity rule allows the system to repair or recover from perturbations in its synapses as shown by simulations in *Figure 5*. However, this learning rule only applies to the weights that ensure stable, self-sustaining activity in the network. The network connectivity responsible for correctly integrating angular velocity inputs (given by the PEN to EPG connections in the fly) might require more elements than a purely Hebbian rule (*Stringer et al., 2002*), such as the addition of a predictive component (*Vafidis et al., 2022*).

## Evolution of the eight-column circuit

Having derived and experimentally validated the theoretical circuit, we now address another question: whether there might be a reason that insect head direction circuits have an eight-column architecture (*Honkanen et al., 2019*; *Stone et al., 2017*; *Pisokas et al., 2020*). The derivations leading to *Equation 17* are valid for other values of $N$, so there is no a priori reason to expect $N = 8$.

Our reasoning follows recent studies in genetics (*Johnston et al., 2022*; *Dingle et al., 2018*), which argue that an observed organism is more likely to have resulted from a simple than a more complex genome: evolution favours simplicity. We note that powers of two are easier to generate with replication dynamics than other numbers, because they just require each cell to divide a set number of times. Other numbers require that, at some point, two cells resulting from a division must behave differently, necessitating more complex signalling mechanisms and rendering this possibility less likely to have been developed by evolution without some other driving factor. We therefore expect $N$ to be a power of two unless required to be otherwise.

As we show in section 'Circuits with different neuron counts', not all numbers of neurons enable a working circuit. The circuits for $N = 2$ and $N = 4$ are degenerate – either producing a single dimensional encoding, or two disconnected circuits that do not enforce the required circular topology. $N = 8$ is the smallest power of two that could result in a non-degenerate circuit. This hints at the possibility that the eight-column architecture is not a chance evolutionary artefact, but rather that it is the genetically simplest circuit capable of performing heading integration.

## Discussion

In this work we derived an optimal noise-minimising circuit for encoding heading and verified that this circuit matches experimental data from insects. Furthermore, we showed that such a circuit can be developed and maintained by a biological learning rule, and proposed a mathematical argument for the eight-column structure found in insect compass circuits. In this section, we discuss the implications and limitations of these contributions, and outline potential future work.

Heading integration circuits in insects have been extensively studied, with models ranging in complexity from simplified conceptual networks (*Wittmann and Schwegler, 1995*; *Cope et al., 2017*) to sophisticated models constrained by biological data and featuring multiple neuron types (*Kakaria and de Bivort, 2017*; *Su et al., 2017*; *Kim et al., 2017*; *Pisokas et al., 2020*; *Lyu et al., 2022*). Previous theoretical work has argued that a sinusoidal activity encoding is such a common motif in insect navigation because it facilitates elementwise vector addition (*Wittmann and Schwegler, 1995*). However, this cannot be the only reason because as we show there is a whole family of circuits with different encoding patterns that enable easy vector addition. By showing that sinusoidal activity emerges from the theoretically most noise-resilient heading integration circuit, and verifying that the corresponding circuit matches experimental data, we close this explanatory gap.

We also show that our proposed circuit can be developed by a simple Hebbian-based learning rule, and that the presence of the eight-column structure can be explained from the perspective of replication dynamics. Both results align with the idea that evolution should be biased towards structures that are easier to encode in the genome (*Johnston et al., 2022*; *Alon, 2007*) and learn (*Zador, 2019*). To the best of our knowledge, this is the first time that such arguments have been put forward in the context of a specific circuit with a specific function.

Our work still has some unaddressed limitations, in particular regarding the topology of the activity. The use of a circular topology to encode the head direction of the insect is valid only in 2D environments. But many species, including the fruit fly, can navigate in a 3D environment. We argue that even if these insects live in a 3D environment, the third dimension (up-down) is different from the other two

due to gravity and the existence of a hard boundary (the ground). Further studies would be required to investigate the full effects of 3D motion.

We could also investigate circuits that integrate position, not only heading, which would require the activity to have a 2D plane topology instead of a circle. This would be particularly relevant for foraging insects such as bees or ants, whose ability to remember their position with respect to their nest has been the subject of many experimental and computational studies (*Collett, 2019*; *Wehner and Srinivasan, 2003*). The position integrating neurons in these insects are also predicted to have sinusoidal activation patterns (*Wittmann and Schwegler, 1995*; *Vickerstaff and Di Paolo, 2005*; *Stone et al., 2017*).

Finally, another interesting avenue for future work is to compare the encoding of direction in insects with that of mammals, which encode heading in a fundamentally different way that uses many more neurons. This raises a critical question: why would the circuit and encoding be different if navigation follows similar principles across species? We speculate that the difference might lie in the type of navigation that the two classes of animals use. Insects often rely on a reference system that is globally anchored to a certain point or phenomenon, whether it is their nests for ants and bees, the sunlight polarisation pattern for locusts, or the milky way for dung beetles. On the other hand, mammals such as rodents typically do not use global cues but rely on local landmarks that are context-dependent and only occur in specific locations. Therefore, mammals must employ a flexible encoding that can be updated as different environments are explored. Further investigation of this possibility would require a different set of principles than those selected here.

## Materials and methods
### Path integration dynamics for heading and position
Here, we make more precise statements about the dynamics of the circuit. The topology of the activity is a circular line attractor, so that any perturbation falls back into a circle and the position of the activity around the circle represents an angle. In our circuit with the dynamics from *Equation 1*, this circular line attractor is achieved by setting

$$\|\phi' \left[ \mathbf{W}\mathbf{a} \right] \| = 1 \quad \forall \|\mathbf{a}\| = r$$
$$\|\phi' \left[ \mathbf{W}\mathbf{a} \right] \| < 1 \quad \forall \|\mathbf{a}\| > r \tag{22}$$
$$\|\phi' \left[ \mathbf{W}\mathbf{a} \right] \| > 1 \quad \forall \|\mathbf{a}\| < r$$

where $r$ is the radius of the attractor. Given that the dynamics are effectively linear we can apply a Fourier transform and write the original dynamics around the line attractor directly in the spatial Fourier basis,

$$\mathcal{F}_f[\dot{\mathbf{a}}] = \mathcal{F}_f[-\mathbf{a} + \omega * \mathbf{a} + \mathbf{u}(t)], \tag{23}$$

which we can expand to obtain

$$\mathcal{F}_f[\dot{\mathbf{a}}] = -\mathcal{F}_f[\mathbf{a}] + \mathcal{F}_f[\omega]\mathcal{F}_f[\mathbf{a}] + \mathcal{F}_f[\mathbf{u}(t)], \tag{24}$$

which only applies to the circular activity where the dynamics are linear. To incorporate the dynamics that force the activity to return to the cycle, we add a new term into the network dynamics which fulfils *Equation 22* – increasing $\|\mathcal{F}_f[\mathbf{a}]\|$ if $\|\mathbf{a}\| < r$ and decreasing it if $\|\mathbf{a}\| > r$. This increase or decrease can be incorporated as a simple scaling factor on the activity decay term

$$\mathcal{F}_f[\dot{\mathbf{a}}] = -\alpha(r - \|\mathbf{a}\|)\mathcal{F}_f[\mathbf{a}] + \mathcal{F}_f[\omega]\mathcal{F}_f[\mathbf{a}] + \mathcal{F}_f[\mathbf{u}(t)], \tag{25}$$

where $\alpha(r - \|\mathbf{a}\|)$ is a nonlinear smooth function with a single minimum and $\alpha(0) = 1$, which forces the activity to have the appropriate magnitude, $\|\mathbf{a}\| = r$.

Note that this doesn't require the neurons to have access to a global activity magnitude signal because each neuron receives a sufficient number of inputs to locally compute the total activity in the network. See the $f = 1$ circuit in *Figure 3*: each neuron receives inputs from all others except those tuned to an orthogonal direction, but the activity of these orthogonal neurons can be computed from the correlated neurons next to them. For example, neuron 1 receives input from all neurons except

3 and 7, but the activities of these neurons can be computed from the activities of neurons 2 and 4 or 6 and 8, respectively. Note also that this is not the case for degenerate circuits such as $f = 2$ in *Figure 3*, which would require an activation function with access to global information to constrain the total activity.

As the activity always falls back to the circular line attractor, the heading integration is linear around the circle. This implies that any small movement of the animal or the perception of a sensory cue is first projected onto the circle, then linearly integrated. We therefore only consider how network inputs cause the activity to rotate around the network,

$$\mathcal{F}_f[\dot{\mathbf{a}}] = -\alpha(r - \|\mathbf{a}\|)\mathcal{F}_f[\mathbf{a}] + \mathcal{F}_f[\omega]\mathcal{F}_f[\mathbf{a}] + \mathcal{F}_f[\mathbf{u}(t)]_{\perp\mathbf{a}}, \tag{26}$$

where $\mathcal{F}_f[\mathbf{u}(t)]_{\perp\mathbf{a}}$ is the projection of the input signal perpendicular to the current activity pattern.

In more intuitive terms, the neurons have a saturating nonlinear activation function where they modulate their gain based on the total activity in the network. If the activity in the network is above the desired level, $r$, the gain is reduced and the activity decreases, and when the activity of the network is less than desired level, both the gain and the activity increase. Note that in this scenario transient deviations from the line attractor, which would induce nonlinear behaviour in the circuit dynamics, are tolerable. External inputs, $\mathbf{u}(t)$, could transiently modify the shape of the activity, producing activity shapes deviating from what the linear model can accommodate. For example, the shape of the bump attractor could be modified through nonlinearities while the insect attains high angular velocity (*Turner-Evans et al., 2017*).

Such nonlinear dynamics do not conflict with the theory developed here, which only requires linearity when the activity is projected onto the circular line attractor. In our framework, the linearity of integration *at the circular line attractor* is not a computational assumption, but rather it emerges from the principle of symmetry.

## Ambiguities in multiple harmonics decoding with drift

We consider the case where multiple harmonics are used, and their phases have drifted from each other. We only focus on angular drift, rather than noise in the full activity, because as noted in section 'Path integration dynamics for heading and position', any deviations in the overall activity level in the network will dissipate.

For example, we consider the harmonics $f_1$ and $f_2$. The activity is given by *Equation 5*

$$\mathbf{a}_n(\theta) = \cos\left(\frac{2\pi f_1 n}{N} + f_1\theta\right) + \cos\left(\frac{2\pi f_2 n}{N} + f_2\theta\right). \tag{27}$$

If the phase of the second harmonic drifts by $\delta\theta$,

$$\mathbf{a}_n^{\delta\theta}(\theta) = \cos\left(\frac{2\pi f_1 n}{N} + f_1\theta\right) + \cos\left(\frac{2\pi f_2 n}{N} + f_2\theta + \delta\theta\right). \tag{28}$$

We can calculate the alignment between the activity with drift and the activity without drift as a dot product of the activity vectors,

$$\frac{\langle \mathbf{a}(\theta), \mathbf{a}^{\delta\theta}(\theta)\rangle}{\|\mathbf{a}(\theta)\|\|\mathbf{a}^{\delta\theta}(\theta)\|} = \frac{1}{\|\mathbf{a}(\theta)\|^2}\sum_n \mathbf{a}_n(\theta)\mathbf{a}_n^{\delta\theta}(\theta) = 1 + \cos(\delta\theta). \tag{29}$$

However, there are other angles where the alignment is better. For example, we can consider an estimate angle, $\hat{\theta}$, with its corresponding activity, $\mathbf{a}(\hat{\theta})$, which gives

$$\frac{\langle \mathbf{a}(\hat{\theta}), \mathbf{a}^{\delta\theta}(\theta)\rangle}{\|\mathbf{a}(\hat{\theta})\|\|\mathbf{a}^{\delta\theta}(\theta)\|} = \frac{1}{\|\mathbf{a}(\theta)\|^2}\sum_n \mathbf{a}_n(\hat{\theta})\mathbf{a}_n^{\delta\theta}(\theta) = \cos(\theta - \hat{\theta}) + \cos(\theta - \hat{\theta} + \delta\theta). \tag{30}$$

For small $\delta\theta$, this is maximised when $\hat{\theta} = \theta + \delta\theta/2$. However, since $\theta$ is circular, if the drift is $\pi$ then there are two possible positions given by $\hat{\theta} = \theta \pm \pi/2$.

This implies that an encoding using multiple harmonics does not necessarily offer a unique decoding in the presence of noise.

**Table 2.** Distribution of neural activity phases for different harmonics.

We computed $fn \mod 8$ for $n = \{1, 2, 3, 4, 5, 6, 7\}$ and $f = \{1, 2, 3, 4, 5, 6, 7, 8\}$ and found that all possible phases appear for any odd number. This happens because the $\mod 8$ operation imposes an abelian group structure, namely $\mathbb{Z}/8$ and any coprime with 8 will be a generator of the whole group. If we use instead $f = 2$ or $f = 6$ we have that we can divide $N = 8$ and $f$ by 2 and we get the abelian group $\mathbb{Z}/4$ which has four elements. The same argument applies to $f = 4$, leaving only two elements, and for $f = 8$ we get a single element.

|       | $n = 0$ | $n = 1$ | $n = 2$ | $n = 3$ | $n = 4$ | $n = 5$ | $n = 6$ | $n = 7$ | $\gcd(8, f)$ |
|-------|---------|---------|---------|---------|---------|---------|---------|---------|--------------|
| $f = 1$ | 0 | 1 | 2 | 3 | 4 | 5 | 6 | 7 | 1 |
| $f = 2$ | 0 | 2 | 4 | 6 | 0 | 2 | 4 | 6 | 2 |
| $f = 3$ | 0 | 3 | 6 | 1 | 4 | 7 | 2 | 5 | 1 |
| $f = 4$ | 0 | 4 | 0 | 4 | 0 | 4 | 0 | 4 | 4 |
| $f = 5$ | 0 | 5 | 2 | 7 | 4 | 1 | 6 | 3 | 1 |
| $f = 6$ | 0 | 6 | 4 | 2 | 0 | 6 | 4 | 2 | 2 |
| $f = 7$ | 0 | 7 | 6 | 5 | 4 | 3 | 2 | 1 | 1 |
| $f = 8$ | 0 | 0 | 0 | 0 | 0 | 0 | 0 | 0 | 8 |

## Equivalent circuits and degeneracies

### Equivalence under permutation

The activity of neurons given by *Equation 5* implies that the preferred angle of neuron $n$ is given by

$$\varphi_n = 2\pi \frac{fn \mod N}{N}, \tag{31}$$

where in our case $N = 8$. *Table 2* shows $nf \mod (N)$ evaluated for all neurons in the network using different harmonics. We notice that for $f = \{1, 3, 5, 7\}$ all the numbers from zero to seven appear, while for $f = \{2, 6\}$ we only get the even numbers, for $f = 4$ we get only zero and four, and for $f = 8$ there is only zero.

The explanation is based on number theory. If $f$ and $N$ have the greatest common divisor $\gcd(N, f) = d$, then $nf \mod N = 0$ for $n = N/d$. This implies that the preferred angle of neuron $n = 0$ is the same as that of $n = N/d$. When $N, f$ are coprime $d = 1$, $n$ goes from 0 to $N - 1$ without repeating any value. However, when $d > 1$, the neuron $n = N/d$ has the same tuning as the neuron $n = 0$, the neuron $n = N/d + 1$ has the same tuning as the neuron $n = 1$ and so on. In other words, the angular tuning of the neurons has a period of $\gcd(N, f)$.

This divides the possible circuits into the following groups:

- $f = \{1, 3, 5, 7\}$ which contain $N = 8$ angular tunings with values $[0, \pi/4, \pi/2, 3\pi/4, \pi, 5\pi/4, 3\pi/2, 7\pi/4]$ because every odd number is coprime with 8 and thus the $\gcd(N, f) = 1$.
- $f = \{2, 6\}$ which cycle through four possible directions $[0, \pi/2, \pi, 3\pi/2]$ because $\frac{N}{\gcd(N, f)} = 4$.
- $f = 4$ which can only represent two possible directions $[0, \pi]$.
- $f = 8$ which can only represent the direction 0.

### Degenerate circuits

Given the groupings presented in the previous subsection, we notice that not all of them allow the encoding of a full range of angles, (0°, 360°). Notably, for $f = 8$ the neurons only encode one angle, and for $f = 4$ only two complementary angles are encoded, 0 and $\pi$. This means that we cannot represent a full circle, because we need two different dimensions to do so, but in both of these cases we can only encode one.

For $f = 2, 6$ we obtain two groups of neurons, and in each group there are 2 neurons that encode two opposing directions, thus it is possible to cover a full circle. However, the circuit is degenerate, as shown in *Figure 3*. In this case there are no connections between even and odd neurons. Thus,

we have two groups of neurons that are disconnected, and each group encodes only one direction (either north-south or east-west). While the encoding could work in principle, the circuit is decoupled, meaning that there is nothing in this circuit that prevents the activity from having activities outside the circular topology: as the two circuits are disconnected, a given value of activity in one group does not restrict the activity in the other.

## Circuits with different neuron counts

We evaluate the viability of circuits with other values of $N$. From findings in the the previous section, we note that choosing $N$ to be a prime number implies that all frequencies will be coprime with $N$, and thus that all the neurons will have a different tuning.

For $N = 2$, the two angles are $0, \pi$, in which case the circuit can only encode one dimension and thus cannot encode a full circle, as was the case for $N = 8, f = 4$ in section 'Degenerate circuits'.

For $N = 4$, if $f = 2$ the circuit can encode only the angles $0, \pi$ and we have the same case as for $N = 2$. But if $f = 1$ or $f = 3$, the neurons are tuned to the angles $0, \pi/2, \pi, 3\pi/2$ which can encode a circle. However, by looking at the connectivity matrix from **Equation 16**, we find that neurons 1, 3 are connected to one another, but not to 2, 4, as was the case for $N = 8, f = 2$ in section 'Degenerate circuits'. This implies that the circuit does not enforce a circular topology, and thus it does not work.

Notice that we could think of another structure where each neuron encodes a different direction. Intuitively, we would have a north-south neuron and an east-west neuron, as we would have in Cartesian coordinates. However, this would have the same problem as $N = 4$, where the 2 neurons would not interact and thus the topology would be wrong. Additionally, when the insect heads north, the north-south neuron would be very active, while when it heads south it would be very inactive. This means that the circuit as a whole would have a different firing rate depending on the direction, breaking the symmetry assumption.

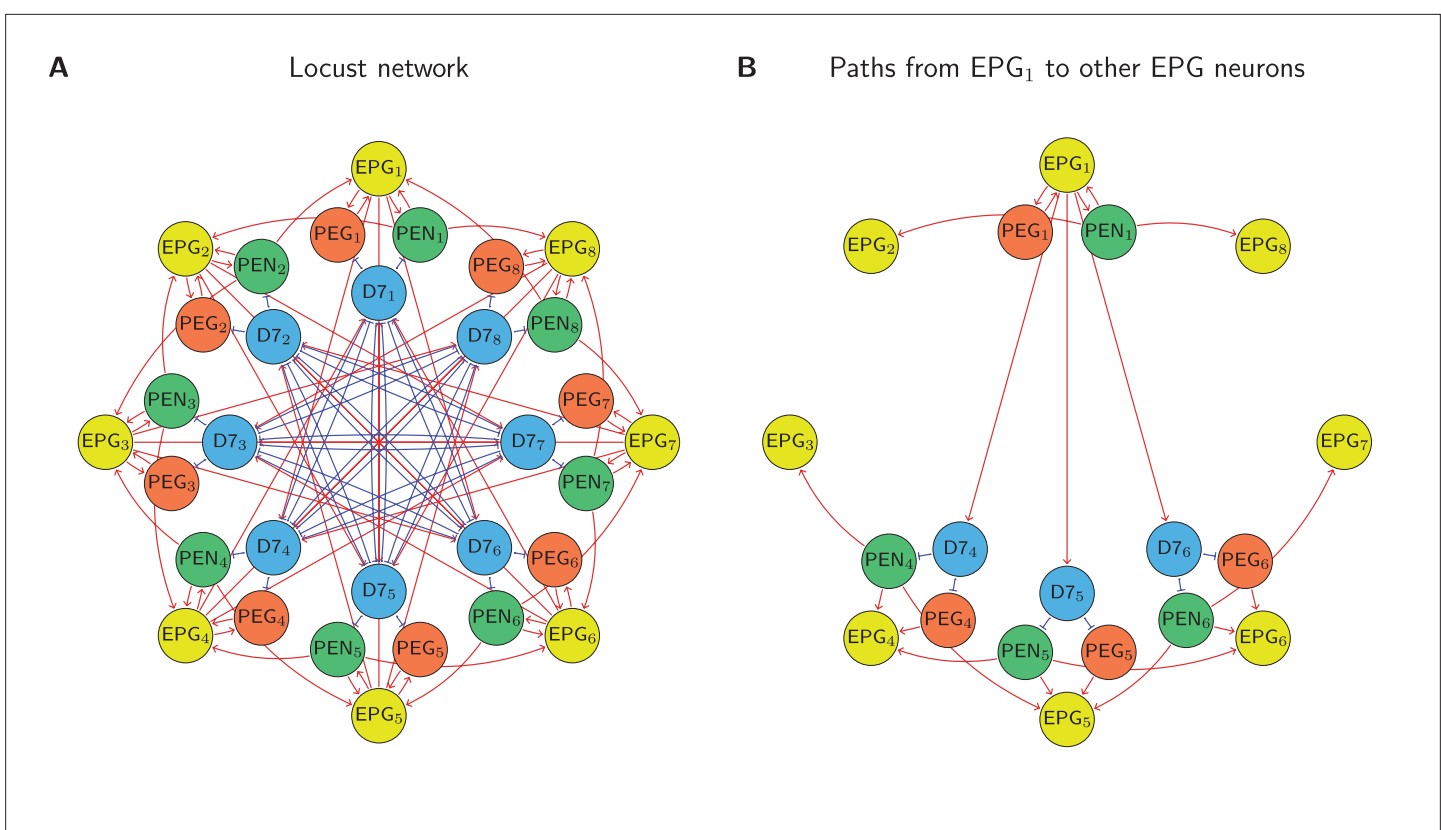

**Figure 6.** Path counting. (**A**) The full locust connectivity network from **Pisokas et al., 2020**, and (**B**) the connections with path lengths 2 and 3 from $EPG_1$ to all other $EPG$ neurons in the network. Because the network is rotationally symmetric these path counts generalise to all $EPG$ neurons as shown in **Table 3**.

**Table 3.** Net path count profile between EPG neurons in the locust circuit.

| $\text{EPG}_n \rightarrow$ | $n-4$ | $n-3$ | $n-2$ | $n-1$ | $n$ | $n+1$ | $n+2$ | $n+3$ |
|---|---|---|---|---|---|---|---|---|
| Excitatory path count | 0 | 0 | 0 | 1 | 2 | 1 | 0 | 0 |
| Inhibitory path count | 4 | 3 | 1 | 0 | 0 | 0 | 1 | 3 |
| Net path count | −4 | -3 | −1 | 1 | 2 | 1 | −1 | −3 |

## Path counting

We counted the number of different paths between EPG neurons, accounting for the sign of the connections (whether the path passed through an inhibitory 7 neuron), and used the net path count as a proxy for connectivity strength. The results of this analysis are shown in section 'Comparing the predicted circuit with biological data', but we will detail the process here using the locust network from *Pisokas et al., 2020*, as an example, shown in *Figure 6*.

We consider the shortest excitatory and inhibitory pathways between EPG neurons, which have lengths of 2 and 3 respectively. There are no direct paths between EPG neurons in the locust circuit. In this case there are two paths of length 2 that implement self-excitatory connections for $\text{EPG}_1$:

- $\text{EPG}_1 \rightarrow \text{PEG}_1 \rightarrow \text{EPG}_1$
- $\text{EPG}_1 \rightarrow \text{PEN}_1 \rightarrow \text{EPG}_1$

And there is one connection of path length 2 that connects $\text{EPG}_1$ to each of its nearest neighbours:

- $\text{EPG}_1 \rightarrow \text{PEN}_1 \rightarrow \text{EPG}_2$
- $\text{EPG}_1 \rightarrow \text{PEN}_1 \rightarrow \text{EPG}_8$

For inhibitory connections there are four paths of length 3 that connect $\text{EPG}_1$ to the neuron on the opposite side of the ring, $\text{EPG}_5$. These are:

- $\text{EPG}_1 \rightarrow \text{Delta7}_5 \rightarrow \text{PEN}_5 \rightarrow \text{EPG}_5$
- $\text{EPG}_1 \rightarrow \text{Delta7}_5 \rightarrow \text{PEG}_5 \rightarrow \text{EPG}_5$
- $\text{EPG}_1 \rightarrow \text{Delta7}_4 \rightarrow \text{PEN}_4 \rightarrow \text{EPG}_5$
- $\text{EPG}_1 \rightarrow \text{Delta7}_6 \rightarrow \text{PEN}_6 \rightarrow \text{EPG}_5$

There are three connections of path length 3 connecting $\text{EPG}_1$ to $\text{EPG}_4$:

- $\text{EPG}_1 \rightarrow \text{Delta7}_4 \rightarrow \text{PEN}_4 \rightarrow \text{EPG}_4$
- $\text{EPG}_1 \rightarrow \text{Delta7}_4 \rightarrow \text{PEG}_4 \rightarrow \text{EPG}_4$
- $\text{EPG}_1 \rightarrow \text{Delta7}_5 \rightarrow \text{PEN}_5 \rightarrow \text{EPG}_4$

Finally, there is one path of length 3 connecting $\text{EPG}_1$ to the neuron perpendicular to it around the ring, $\text{EPG}_3$:

- $\text{EPG}_1 \rightarrow \text{Delta7}_4 \rightarrow \text{PEN}_4 \rightarrow \text{EPG}_3$

We then add these connections together to obtain the net path count profile for the network. Because the network is rotationally symmetric, the connections from $\text{EPG}_1$ are the same as the connections from $\text{EPG}_2$ but with the neurons indices incremented by 1. The generalised profile is shown in *Table 3*.

**Table 4.** Neuron types and their numbers in the fruit fly connectome dataset (*Scheffer et al., 2020*).

| Neuron class | Number |
|---|---|
| PEN_a(PEN1) | 20 neurons |
| PEN_b(PEN2) | 22 neurons |
| PEG | 18 neurons |
| EPG | 46 neurons |
| EPGt | 4 neurons |
| Delta7 (Δ7) | 42 neurons |

## Data preprocessing

The following section details our data preprocessing to produce a central complex circuit for the fruit fly similar to that in *Pisokas et al., 2020*, using synaptic counts from the fruit fly connectome dataset (*Scheffer et al., 2020*). Full details are shown in our available code.

We first identified the 6 neuron types which corresponded to the neurons of interest in the central complex (*Table 4*).

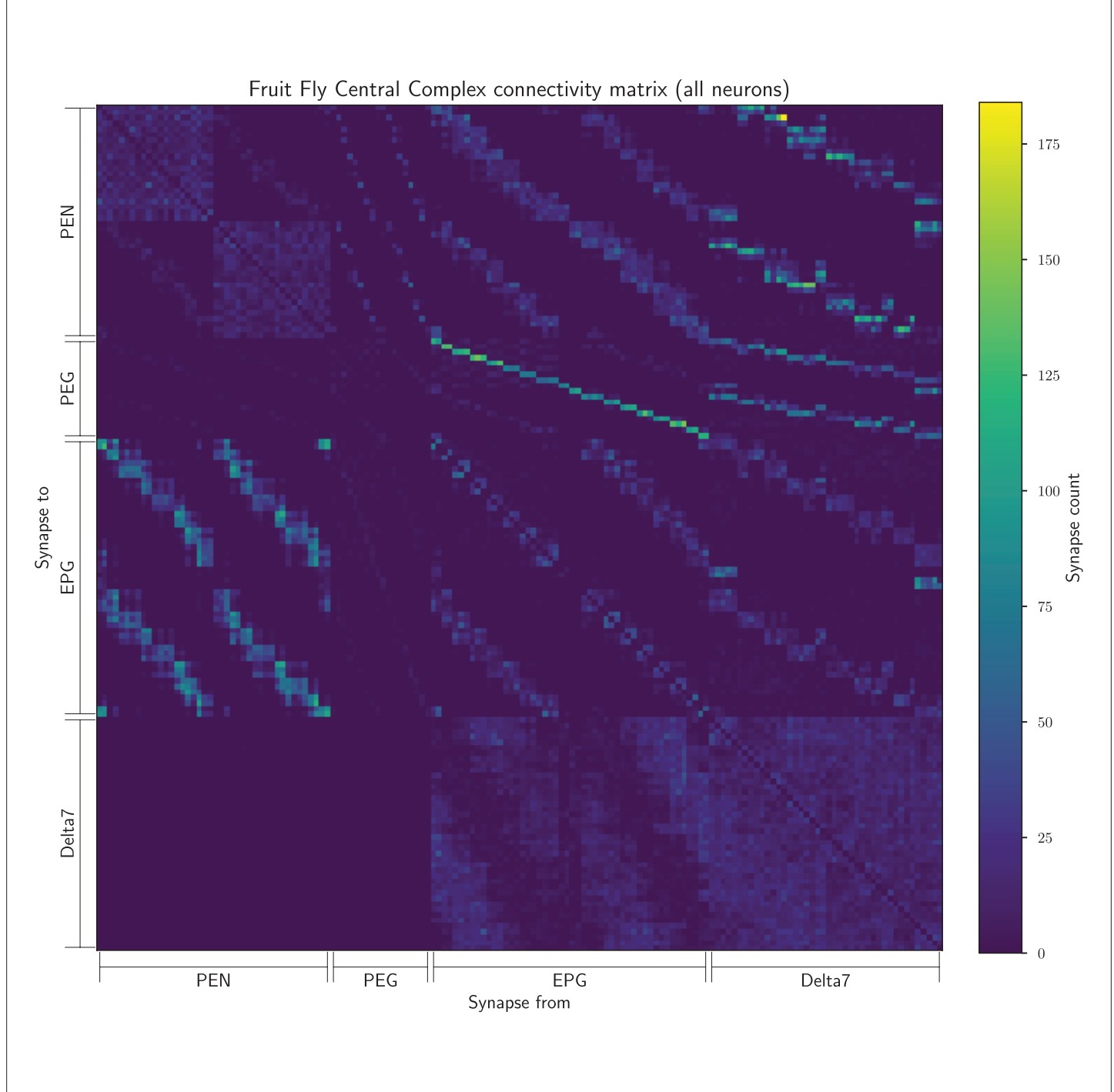

**Figure 7.** The connectivity matrix for the fruit fly containing all 152 neurons. On the horizontal axis are the names of the pre-synaptic neurons while on the vertical axis the names of the post-synaptic neurons. Neurons of each cell type are ordered by the glomerulus they innervate and arbitrarily within glomerulus.

**Table 5.** List of the Delta7 neuron sub-types in the fruit fly connectome dataset (*Scheffer et al., 2020*).

These differ in the glomeruli their pre- and post-synaptic terminals innervate. We are referring to those altogether as Delta7 neurons in the present account.

| Delta7 neuron sub-types | Glomerulus group |
|---|---|
| Delta7(PB15)_L1L9R8_R | Delta7 L1 |
| Delta7(PB15)_L2R7_R | Delta7 L2 |
| Delta7(PB15)_L3R6_R | Delta7 L3 |
| Delta7(PB15)_L4R5_R | Delta7 L4 |
| Delta7(PB15)_L4R6_R | Delta7 L4 |
| Delta7(PB15)_L5R4_L | Delta7 L5 |
| Delta7(PB15)_L6R3_L | Delta7 L6 |
| Delta7(PB15)_L6R4_L | Delta7 L6 |
| Delta7(PB15)_L7R2_L | Delta7 L7 |
| Delta7(PB15)_L7R3_L | Delta7 L7 |
| Delta7(PB15)_L8R1R9_L | Delta7 L8 |

We grouped the two PEN and two EPG populations together for further analysis.

From the connectome data we created a connectivity matrix (*Figure 7*) containing the number of synapses between all pairs of neurons of the above-mentioned types. This contained 129,473 synapses between 152 neurons in total.

Next, neurons in the same glomerulus were grouped together. The EPG neurons were divided between 18 glomeruli, L1-L9 and R1-R9. The PEN neurons were also divided between 16 glomeruli, L2-L9 and R2-R9. The PEG neurons were divided into 18 glomeruli with only 1 neuron in each, L1-L9 and R1-R9. The Delta7 neurons had 10 unique sub-types (*Table 5*) which were grouped into eight glomeruli based on their left glomerulus index – i.e., L4R5_R and L4R6_R were grouped in L4.

After this grouping we had a connectivity matrix of 60 neurons in total (*Figure 8*).

For the EPG, PEN, and PEG neurons we then grouped glomeruli mirrored in the two hemispheres together – i.e., L1 and R1 were grouped into glomerulus 1. This resulted in a connectivity matrix of 34 neurons (*Figure 9*).

Finally, as in *Pisokas et al., 2020*, we grouped the PEG9 and PEG1 neurons and EPG9 and EPG1 neurons together. The former because they both output to the same ellipsoid body segment, and the latter because they both receive common input. This resulted in a connectivity matrix with 32 neurons (*Figure 10*) which we used for the analysis in *Figure 4* as this network had the eightfold rotational symmetry compatible with our theoretical model.

## Fitting weights

Having calculated the paths or synaptic strengths between EPG neurons in the network, we next computed the average connectivity profile for the network – how each EPG neuron connected to its neighbours a certain distance around the ring. We then compared this connectivity profile to the closest fitting sinusoid to quantify to what degree our prediction of sinusoidal weights was consistent with biological data.

We use the vector

$$\hat{\omega}_d = \beta \cos\left(\frac{2\pi d}{N}\right) + \gamma, \tag{32}$$

where $d = ((m - n) + 4 \mod 8) - 4 \in [-4...3]$ is the signed circular distance from neuron $m$ to neuron $n$.

In the networks from *Pisokas et al., 2020*, the weights are rotationally symmetric, so we minimise the mean squared error between the sinusoidal profile and the biological connectivity profile,

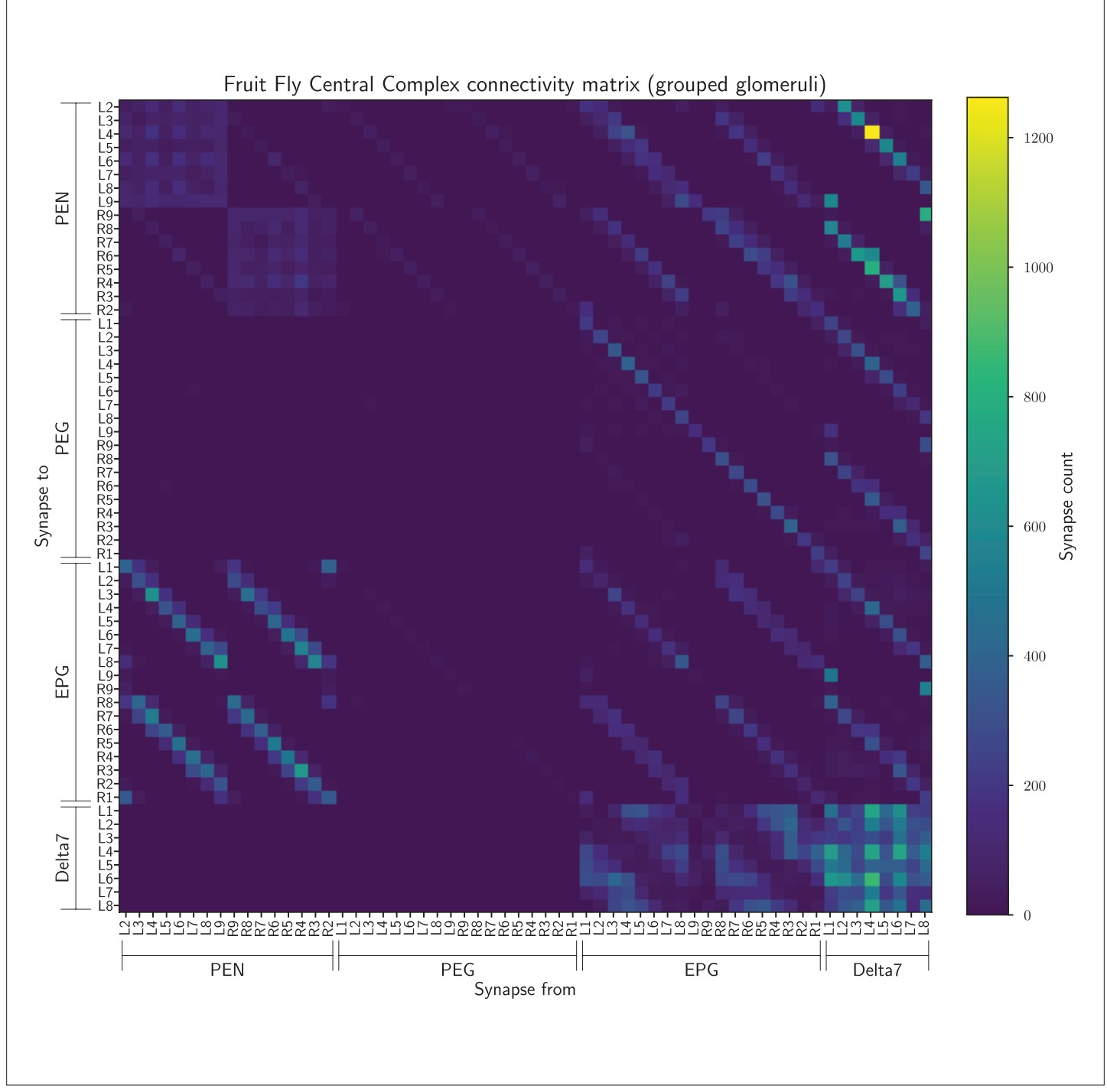

**Figure 8.** The connectivity matrix for the fruit fly neurons grouped by innervated glomerulus so containing 60 groups. On the horizontal axis are the names of the pre-synaptic neurons while on the vertical axis the names of the post-synaptic neurons. Neuron groups are ordered by the glomerulus they innervate.

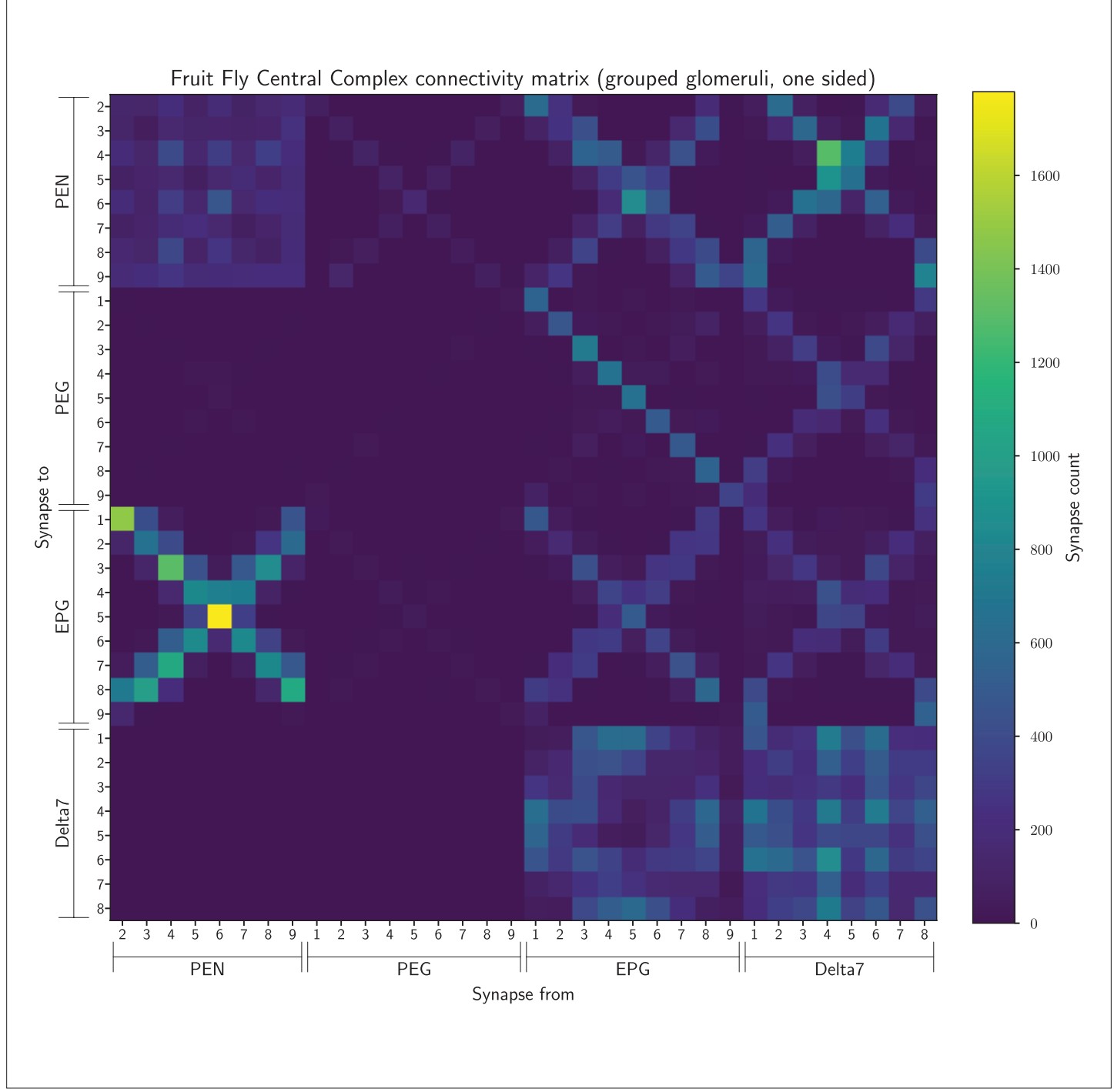

**Figure 9.** The connectivity matrix for the fruit fly neurons grouped by glomerulus and aggregated from both hemispheres, so containing 34 groups. On the horizontal axis are the names of the pre-synaptic neuron groups while on the vertical axis are the names of the post-synaptic neuron groups. Neuron groups are ordered by the glomerulus they innervate.

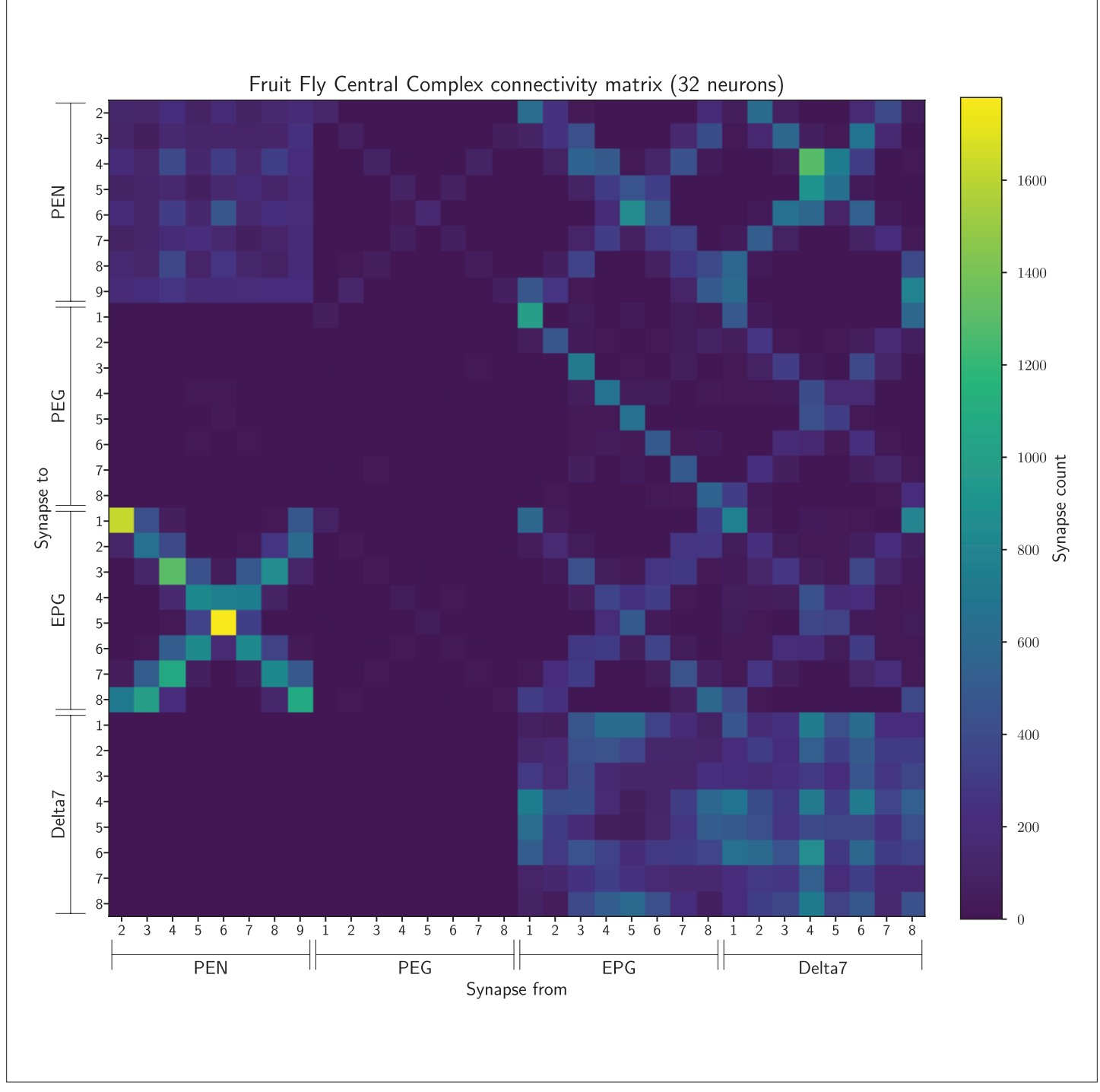

**Figure 10.** The connectivity matrix for the fruit fly neurons grouped by glomerulus and aggregated from both hemispheres, with PEG9 and PEG1 neurons and EPG9 and EGP1 neurons grouped, so containing 32 groups in total. On the horizontal axis are the names of the pre-synaptic neuron groups while on the vertical axis the names of the post-synaptic neuron groups. Neuron groups are ordered by the glomerulus they innervate.

$$\beta^*, \gamma^* = \text{argmin}_{\beta,\gamma} \frac{1}{N} \sum_{d=-N/2}^{N/2-1} (\hat{\omega}_d(\beta,\gamma) - \omega_d)^2, \tag{33}$$

by using least squares.

The network derived from the synaptic count data in *Scheffer et al., 2020*, is not rotationally symmetric so each value in the average connectivity profile, $\omega_d$, has a corresponding standard deviation, $\sigma_d$. In this case we minimise the precision-weighted mean squared error, which emphasises fitting connectivity profile values that are more consistently seen in the network:

$$\beta^*, \gamma^* = \text{argmin}_{\beta,\gamma} \frac{1}{N} \sum_{d=-N/2}^{N/2-1} \left( \frac{\hat{\omega}_d(\beta,\gamma) - \omega_d}{\sigma_d} \right)^2. \tag{34}$$

The result of fitting sinusoidal profiles to the data is shown in section 'Comparing the predicted circuit with biological data'. Since the fit for the fruit fly is not as clear as for the locust, we also compared the observed weights in the fruit fly with Gaussian and von Mises curves. Since both the Gaussian and von Mises distributions also have a parameter specifying their width, this requires fitting an extra parameter. For the Gaussian we have

$$\beta_g e^{-\frac{n^2}{\sigma_g}} + \gamma_g, \tag{35}$$

where $\beta_g, \sigma_g, \gamma_g$ are parameters to fit. For the von Mises distribution we have

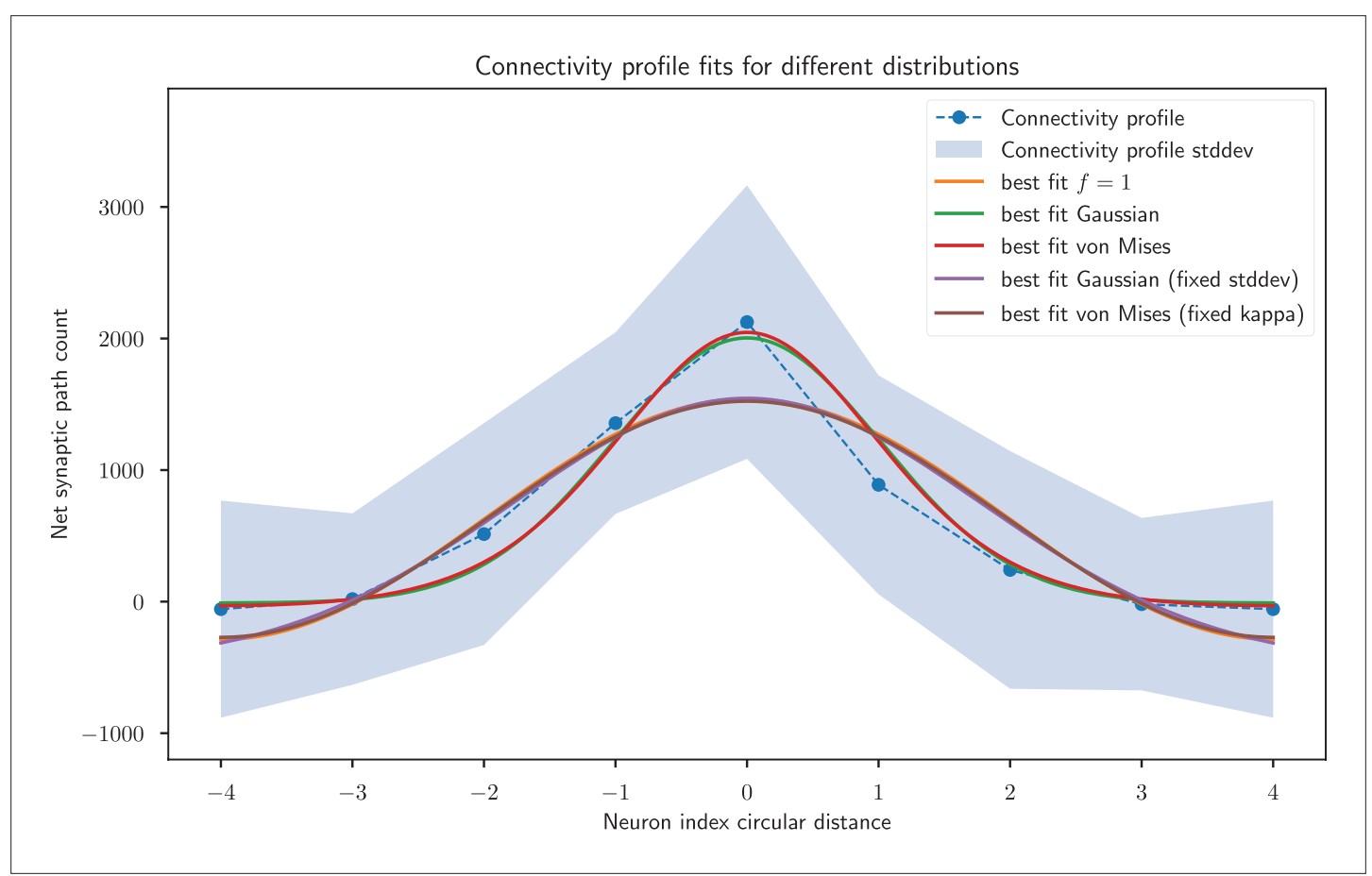

**Figure 11.** Gaussian and von Mises distributions provide a better fit than the weight profile with sinusoidal weights of frequency 1, but the former two distributions have an extra model parameter. When fixing the width parameters of these distributions to match the sinusoid, all three distributions provide a very similar fit. Note that the –4 and 4 neuron indices are the same and just duplicated for visualisation purposes.

**Table 6.** Quality of fit for different possible weight structures.
The root mean square error (RMSE) is weighted by the variance attached to each connection (in units of standard deviations). The corrected Akaike information criterion (AICc) is also shown for the weight profiles in *Figure 11*. The best fit after correcting for model size is the Gaussian with fixed standard deviation, but it is very closely matched by the von Mises with fixed kappa and $f = 1$.

| Curve | RMSE | AICc |
|---|---|---|
| $f = 1$ | 0.3213 | 8.05 |
| Gaussian | 0.1941 | 12.60 |
| von Mises | 0.1854 | 12.55 |
| Gaussian (fixed stddev) | 0.3142 | 7.98 |
| von Mises (fixed kappa) | 0.3178 | 8.02 |

$$\beta_v e^{\kappa_v \cos(n)} + \gamma_v, \tag{36}$$

where $\beta_v, \kappa_v, \gamma_v$ are parameters to fit. For both the Gaussian and von Mises, we find a good agreement with the weights in *Figure 11*, and a lower root mean square error (RMSE) than with any combination of harmonics. However, we note that our sinusoidal model has two parameters instead of three, so to make a fair comparison we use the corrected Akaike information criterion (AICc) (*Burnham and Anderson, 2004*), which is given by

$$\text{AICc} = 2p - 2\ln(L) + \frac{2p^2 + 2p}{N - p - 1} = 2p + 2N\,\text{RMSE}^2 + \frac{2p^2 + 2p}{N - p - 1}, \tag{37}$$

where $p$ is the number of distribution parameters, $N$ the number of samples, and $\ln(L)$ is the log-likelihood, which in this case is the sum of squared errors, $\ln(L) = -N\text{RMSE}^2$. We find that the lowest AICc (corresponding to the best model) corresponds to the harmonic $f = 1$ (*Table 6*).

As a complementary approach to evaluate the shape of the distribution, we first fit the Gaussian and von Mises distributions to the best fit $f = 1$ curve. We then freeze the width parameters of the distributions ($\sigma_g$ for the Gaussian and $\kappa_v$ for the von Mises) and only optimise the amplitude and vertical offset parameters ($\beta$ and $\gamma$) to fit the data. This approach limits the number of free parameters for the Gaussian and von Mises distributions to two, to match the sinusoid. The results are shown in *Figure 11* and *Table 6*. Both the fixed-width Gaussian and von Mises distributions are a slightly better fit to the data than the sinusoid, but the differences between the three curves are very small.

In simplifying the fruit fly connectome data, we assumed all synapses of different types were of equal weight, as no data to the contrary were available. Different synapse types having different strengths could introduce nonlinear distortions between our net synaptic path count and the true synaptic strength, which could in turn make the data a better or worse fit for a sinusoidal compared to a Gaussian profile. As such, we don't consider the 2% relative difference in RMSE between the $f = 1$ sinusoid and fixed-width Gaussian and von Mises distributions to be conclusive.

Overall, we find that the cosine weights that emerge from our derivations are a very close match for the locust, but less precise for the fly, where other functions fit slightly better. Given the limitations in using the currently available data to provide an exact estimate of synaptic strength (for the locust), and due to the high variability of the synaptic count (for the fruit fly), we consider that our theory is compatible with the observed data.

## Convergence of Oja's rule with multiple harmonics

### Linear neurons

Integrating our modified Oja's learning rule updates (*Equation 19*) over all angles in the position space gives us the following:

$$\Delta\omega = \mathbf{a} * \mathbf{a} - \|\mathbf{a}\|^2 \omega, \tag{38}$$

which we can transform into the Fourier domain:

$$\mathcal{F}_f[\Delta\omega] = \mathcal{F}_f[\mathbf{a}]^2 - \|\mathbf{a}\|^2 \mathcal{F}_f[\omega]. \tag{39}$$

The steady-state solution when the weight update is 0 is:

$$\mathcal{F}_f[\omega] = \frac{\mathcal{F}_f[\mathbf{a}]^2}{\|\mathbf{a}\|^2}. \tag{40}$$

By Parseval's theorem, $\|\mathbf{a}\|^2 = \sum_f \|\mathcal{F}_f[\mathbf{a}]\|^2$ so the Fourier spectrum of the steady-state weights is just the normalised Fourier spectrum of the input:

$$\mathcal{F}_f[\omega] = \frac{\mathcal{F}_f[\mathbf{a}]^2}{\sum_f \|\mathcal{F}_f[\mathbf{a}]\|^2}. \tag{41}$$

From this we can see that the stable $L_1$ norm of $\mathcal{F}[\omega]$ is 1,

$$\sum_f \left|\mathcal{F}_f[\omega]\right| = \frac{1}{\sum_f \|\mathcal{F}_f[\mathbf{a}]\|^2} \sum_f \left|\mathcal{F}_f[\mathbf{a}]^2\right| = 1. \tag{42}$$

If we combine this result with **Equation 10**, we find that in the case of a single encoding frequency, $\mathcal{F}_{f*}[\mathbf{a}] > 0$, Oja's rule will result in a single stable harmonic with $\mathcal{F}_{f*}[\omega] = 1$ because $\sum_f \mathcal{F}_f[\mathbf{a}]^2 = \mathcal{F}_{f*}[\mathbf{a}]^2$.

If Oja's rule has a normalising factor added to account for the number of encoding harmonics used, $|\mathbf{F}|$,

$$\Delta\omega = \mathbf{a} * \mathbf{a} - \frac{1}{|\mathbf{F}|}\|\mathbf{a}\|^2 \omega, \tag{43}$$

then the steady-state solution for the weights becomes

$$\mathcal{F}_f[\omega] = \left|\mathbf{F}\right| \frac{\mathcal{F}_f[\mathbf{a}]^2}{\sum_f \mathcal{F}_f[\mathbf{a}]^2}, \tag{44}$$

where the $L_1$ norm of $\omega = |\mathbf{F}|$. In this case multiple harmonics could develop stable values of $\mathcal{F}_f[\omega] = 1$, but only if the activity magnitudes for these harmonics are identical. If any perturbation affects the activities, the harmonic which happens to have the larger activity will begin to dominate the other by the dynamics of **Equation 43**, until only one remains.

Therefore, the only case where our learning rule results in a stable solution robust to perturbations is when only one harmonic is used.

## Neurons with nonlinear activations

An analysis similar to the previous case applies when the firing rate of the neurons has a nonlinear activation. For a given activity profile, **Equation 41** still applies. Thus, for an activity profile $\mathbf{a}$, it will generate the weight profile

$$\omega = \frac{1}{\|\mathbf{a}\|^2} \sum_f \left|\mathcal{F}_f\left[\mathbf{a}\right]\right|^2, \tag{45}$$

where we set $\|\omega\| = 1$ for convenience of notation. Note that the squaring process will induce a weight distribution in which the larger harmonic will be enhanced more than the rest.

Having obtained the weights, the equation for the activity profile can be solved through the self-consistency setting in which $u(t) = 0$ and the activity is maintained,

$$\mathbf{a}_n(\theta) = \phi\left(\sum_{m=0}^{N} \omega_m \mathbf{a}_{n-m}(\theta)\right), \tag{46}$$

where $\phi$ is the nonlinear activation function. Given the rotational symmetry from our assumptions, it is enough to solve for $\mathbf{a}_n = \mathbf{a}_n(0)$, which gives us a set of $N/2$ equations which can then be combined with **Equation 45**, which gives us the following equation to solve

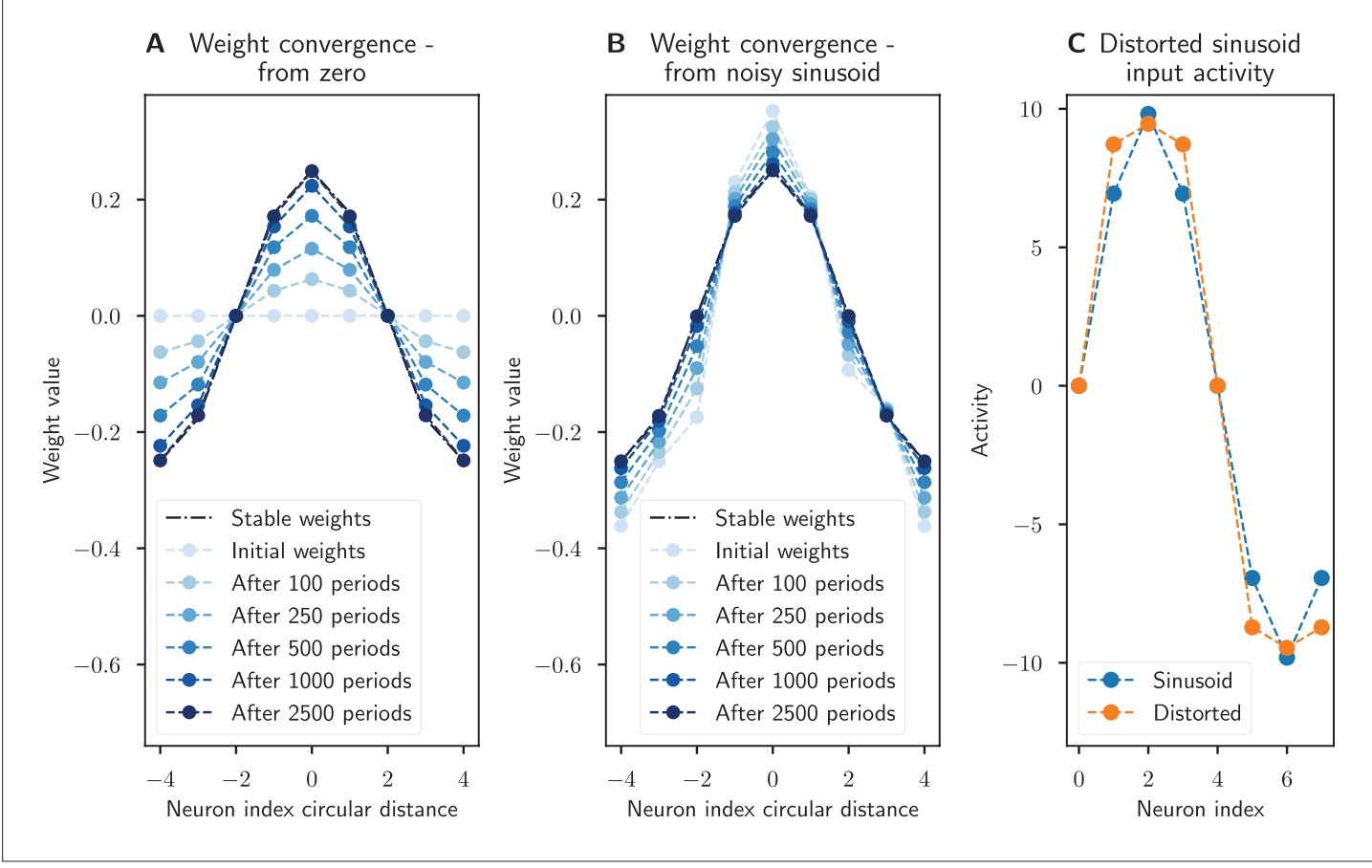

**Figure 12.** Recovery of synaptic weights with a Hebbian learning rule. The synaptic weights converge to a sinusoidal pattern under Oja's rule when the neurons have a nonlinear (tanh) activation function and a distorted (more square wave-like) sinusoidal input is provided. (**A**) The weights start at zero and slowly converge to the prescribed sinusoidal profile, showing that this connectivity can emerge from scratch. (**B**) The sinusoidal weights are perturbed by noise but learning ensures that the weight profile is corrected. (**C**) The distorted sinusoidal input (orange) compared to a sinusoid profile (blue). Noise was added to the initial activity, weights, and input with the same parameters as in *Figure 5*, the only difference being that these simulations were run for 2500 periods.

$$\mathbf{a}_n = \phi\left(\sum_{m=0}^{N}\left(\sum_{f=1}^{N}\frac{\mathcal{F}_f[\mathbf{a}]^2}{\|\mathbf{a}\|^2}\cos\left(\frac{2\pi(n-m)}{N}\right)\right)\mathbf{a}_{n-m}\right)$$

$$= \phi\left(\frac{1}{\|\mathbf{a}\|^2}\sum_{m=0}^{N}\left(\sum_{f=1}^{N}\left[\sum_{k=0}^{N}\mathbf{a}_k\cos\left(\frac{2\pi kf}{N}\right)\right]^2\cos\left(\frac{2\pi(n-m)}{N}\right)\right)\mathbf{a}_{n-m}\right).$$

(47)

Since $\mathbf{a}_n = \mathbf{a}_{-n \mod N}$, we obtain $N/2$ equations for $N/2$ values of $\mathbf{a}_n$.

Note that there is no closed form solution in general, but some generic properties can be inferred. For example, as long as we have a single bump of activity, $\mathcal{F}_1[\mathbf{a}] > |\mathcal{F}_f[\mathbf{a}]|$, $\forall f > 1$, therefore $\mathcal{F}_1[\omega] > \mathcal{F}_f[\omega]$, $\forall f > 1$. More specifically,

$$\mathcal{F}_f[\omega] = \frac{|\mathcal{F}_f[\mathbf{a}]|^2}{|\mathcal{F}_1[\mathbf{a}]|^2}\mathcal{F}_1[\omega].$$

(48)

*Figure 12* shows the results of the same simulation as section 'Learning rules and development', but using neurons with a nonlinear (tanh) activation function, and distorting the network input to be a more square wave-like sinusoid. As predicted theoretically, our rule still reinforces the dominant spatial frequency in the network, causing the weights to converge to the sinusoidal profile our theory predicts.

## Acknowledgements

We would like to thank Benjamin F Grewe for his support and helpful comments.

## Additional information

### Funding

| Funder | Grant reference number | Author |
|---|---|---|
| Royal Society of Edinburgh | Saltire Early Career Fellowship | Ioannis Pisokas |
| Swiss National Science Foundation | Project Grant 182539 | Pau Vilimelis Aceituno |

The funders had no role in study design, data collection and interpretation, or the decision to submit the work for publication.

### Author contributions

Pau Vilimelis Aceituno, Ioannis Pisokas, Conceptualization, Formal analysis, Supervision, Validation, Investigation, Methodology, Writing – original draft, Project administration, Writing – review and editing; Dominic Dall'Osto, Software, Formal analysis, Validation, Investigation, Visualization, Methodology, Writing – original draft, Writing – review and editing

### Author ORCIDs

Pau Vilimelis Aceituno (iD) https://orcid.org/0000-0002-1218-4009
Dominic Dall'Osto (iD) http://orcid.org/0000-0002-9549-4490
Ioannis Pisokas (iD) https://orcid.org/0000-0001-7426-3207

### Decision letter and Author response

Decision letter https://doi.org/10.7554/eLife.91533.sa1
Author response https://doi.org/10.7554/eLife.91533.sa2

## Additional files

### Supplementary files

• MDAR checklist

### Data availability

The current manuscript is a computational study, so no data have been generated for this manuscript. All modelling and analysis code is available at GitHub (copy archived at *Dall'Osto, 2024*).

The following previously published dataset was used:

| Author(s) | Year | Dataset title | Dataset URL | Database and Identifier |
|---|---|---|---|---|
| Xu CS, Scheffer L, Plaza S, Januszewski M, Lu Z, Takemura SY, Hayworth K, Shinomiya K, Maitin-Shepard J, Bogovic J, Hubbard P, Kainmueller D, Katz W, Li PH, Neubarth N, Schlegel P, Neace ER, Knecht CJ, Alvarado CX, Bailey D, Ballinger S, Borycz JA, Canino B, Cook M, Duclos O, Eubanks B, Finley S, Forknall N, Francis A, Hopkins GP, Joyce EE, Kim S, Kirk NA, Kovalyak J, Lauchie SA, Lohff A, Maldonado C, Manley EA, McLin S, Mooney C, Ndama M, Ogundeyi O, Okeoma N, Ordish C, Padilla N, Patrick C, Phillips EE, Phillips EM, Rampally N, Ribeiro C, Robertson MK, Rymer JT, Ryan SM, Sammons M, Scott AK, Scott AL, Shinomiya A, Smith C, Smith NL, Sobeski MA, Suleiman A, Takemura S, Talebi R, Tarnogorska D, Tenshaw E, Tokhi T, Walsh JJ, Horne JA, Parekh R, Rivlin PK, Jayaraman V, Meinertzhagen I, Rubin GM, Jain V | 2020 | Data for A Connectome of the Adult *Drosophila* Central Brain v1.0 | https://doi.org/10.25378/janelia.11676099 | Janelia Research Campus, 10.25378/janelia.11676099 |

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
