## [Editor Report]

This important work suggests that the observed cosine-like activity in the head direction circuit of insects not only subserves vector addition but also minimizes noise in the representation. The authors provide solid evidence using the locust and fruit fly connectomes. The work raises important theoretical questions about the organization of the navigation system and will be of interest to theoretical and experimental researchers studying navigation.

---

## [Decision Letter]

**Decision letter after peer review:**

Thank you for submitting your article "The insect compass system: From theory to circuitry" for consideration by *eLife*. Your article has been reviewed by 1 peer reviewer, and the evaluation has been overseen by a Reviewing Editor and Laura Colgin as the Senior Editor.

The reviewer and the Senior Editor have discussed your submission, and the Senior Editor has drafted this to help you prepare a revised submission.

Thank you for your patience. It was difficult to find reviewers for your manuscript. Then, one of the reviewers did not submit their review and did not respond to emails over the course of multiple months. In fairness to you, we decided to move forward with the decision on the basis of one review.

Essential revisions (for the authors):

*Reviewer #2 (Recommendations for the authors):*

– L. 186 "..these extra encoding channels require a higher total neural activity…" A sentence explaining why this is the case would be helpful to the reader. For example, this might be because we require a localized bump of activity.

– L. 18 "finding a remarkable agreement between our theory and experimental evidence", L. 267 "We therefore confirm our prediction…" I believe the language here should be toned down (see weaknesses point 2).

– Some explanation of how eq. 23 is derived would be beneficial.

­– Methods section 3.2 is instructive but a sentence explaining the choice to model the impact of noise as a phase drift δθ would be beneficial. ­

– L. 427 the degeneracy argument holds if the weights between the two sets of neurons are zero. Another argument against this circuit is that it has half the spatial resolution.

– L. 297-9 This statement seems a bit harsh. Picking up the dominant frequency would still be a great result.

– L. 342-4 The idea to learn the HD system by simple Hebbian learning rules has been proposed before by Stringer et al. 2002, and potentially others. The authors should cite this work.

---

## [Author Response]

Essential revisions (for the authors):– L. 186 "..these extra encoding channels require a higher total neural activity…" A sentence explaining why this is the case would be helpful to the reader. For example, this might be because we require a localized bump of activity.

We added the appropriate explanation (lines 184-188).

“As mentioned, the signal-to-noise ratio in the network remains constant as additional encoding channels, c, are used. But additional channels imply additional harmonics, which by Parseval's theorem require a higher total neural activity.”

– L. 18 "finding a remarkable agreement between our theory and experimental evidence", L. 267 "We therefore confirm our prediction…" I believe the language here should be toned down (see weaknesses point 2).

We modified the text accordingly, and now state that our theory is consistent with the data (line 15 and lines 268-274).

“We repeated the path counting analysis for the fruit fly with synapse count data (Figure 4, B), and found that while the data are noisy, the connectivity profile fits a single sinusoid pattern reasonably well. However, the high variability in the synapse counts makes our hypothesis difficult to differentiate from alternative shapes (see Section 4.6).

Taken together, this analysis shows that our theory is consistent with experimental data – using connectivity-level data for the desert locust and synapse-count data for the fruit fly.”

– Some explanation of how eq. 23 is derived would be beneficial.

We added an explanation (lines 396-399)

“To incorporate the dynamics that force the activity to return to the cycle, we add a new term into the network dynamics which fulﬁls Eq. 22 – increasing ‖F_f[a]‖ if ‖a‖< r and decreasing it if ‖a‖> r. This increase or decrease can be incorporated as a simple scaling factor on the activity decay term”

– Methods section 3.2 is instructive but a sentence explaining the choice to model the impact of noise as a phase drift δθ would be beneficial.

We added a sentence explaining that given the circular line attractor dynamics – which emerge from our initial assumptions – any perturbation would eventually be equivalent to a phase drift (lines 422-423).

“We only focus on angular drift, rather than noise in the full activity, because as noted in Section 4.1, any deviations in the overall activity level in the network will dissipate.”

– L. 427 the degeneracy argument holds if the weights between the two sets of neurons are zero. Another argument against this circuit is that it has half the spatial resolution.

We agree with the statement about the degeneracy, and note that the zero weights would also emerge from a learning rule if the neurons are uncorrelated with zero correlations. The notion of spatial resolution is perhaps slightly confusing here. But in terms of “decoding accuracy” – i.e. how much the represented angle is corrupted by the addition of zero mean noise to each neuron’s activity – the circuits are equivalent. Even though the f=2 circuit only represents half the number of directions as the f=1 circuit, each direction has half the noise variance because the two neurons’ activities can be averaged.

– L. 297-9 This statement seems a bit harsh. Picking up the dominant frequency would still be a great result.

The dominant frequency is indeed amplified with respect to the others. We emphasised this point (lines 307-312).

“An important point in the use of Oja’s rule is that it would tend to concentrate the activity in a single harmonic. In a linear network, the harmonics would compete during learning, leading to one single harmonic emerging and all others being suppressed, as shown in Section 4.7.1. For neurons with a nonlinear activation function, secondary harmonics would emerge, but would remain small under mild assumptions, as shown in Section 4.7.2. Oja’s rule will still cause the weights to converge to approximately sinusoidal connectivity.”

– L. 342-4 The idea to learn the HD system by simple Hebbian learning rules has been proposed before by Stringer et al. 2002, and potentially others. The authors should cite this work..

We have added this reference (see response to major point 3).